# Humid heat environment causes anxiety-like disorder via impairing gut microbiota and bile acid metabolism in mice

Huandi Weng[1,2,3,7], Li Deng[3,7], Tianyuan Wang[3,7], Huachong Xu[3], Jialin Wu[3], Qinji Zhou [3], Lingtai Yu[2], Boli Chen[2], Li'an Huang[1], Yibo Qu [2], Libing Zhou [1,2,4,5,6] ✉ & Xiaoyin Chen [3] ✉

Climate and environmental changes threaten human mental health, but the impacts of specific environmental conditions on neuropsychiatric disorders remain largely unclear. Here, we show the impact of a humid heat environment on the brain and the gut microbiota using a conditioned housing male mouse model. We demonstrate that a humid heat environment can cause anxiety-like behaviour in male mice. Microbial 16 S rRNA sequencing analysis reveals that a humid heat environment caused gut microbiota dysbiosis (e.g., decreased abundance of *Lactobacillus murinus*), and metabolomics reveals an increase in serum levels of secondary bile acids (e.g., lithocholic acid). Moreover, increased neuroinflammation is indicated by the elevated expression of proinflammatory cytokines in the serum and cortex, activated PI3K/AKT/NF-κB signalling and a microglial response in the cortex. Strikingly, transplantation of the microbiota from mice reared in a humid heat environment readily recapitulates these abnormalities in germ-free mice, and these abnormalities are markedly reversed by *Lactobacillus murinus* administration. Human samples collected during the humid heat season also show a decrease in *Lactobacillus murinus* abundance and an increase in the serum lithocholic acid concentration. In conclusion, gut microbiota dysbiosis induced by a humid heat environment drives the progression of anxiety disorders by impairing bile acid metabolism and enhancing neuroinflammation, and probiotic administration is a potential therapeutic strategy for these disorders.

Mental health is a global public concern, and the number of patients with neuropsychiatric disorders is regularly increasing. Anxiety disorders are the most prevalent neuropsychiatric disorders in humans, with a global prevalence rate of 0.9–28.3%[1], and have become the sixth leading cause of disability in both high- and low-income countries[2]. Patients with anxiety disorders often have other neuropsychiatric disorders, such as depression, and functional impairment of emotional and behavioural symptoms may occur across the lifespan[3]. The aetiology of anxiety disorders is complicated, and recent studies have

reported that several risk factors, such as age, sex, genetics, and social stress, are associated with the incidence of anxiety disorders[4].

Climatic factors, including humidity, temperature and air pressure, are significantly correlated with the onset of negative emotions such as fatigue syndrome, depression and anxiety[5]. Previous studies have also shown that extremely high temperature and high humidity can affect emotional and behavioural disorders and increase the number of patients with conditions such as schizophrenia, mania, and neurological disorders[6]. As the problem of global warming further

intensifies, the impact of extremely humid heat environments on negative emotions (e.g., anxiety) in humans will increase further, and the impact of environmental changes on mental health has attracted increasing attention[7,8]. How climate (e.g., temperature and humidity) changes lead to neuropsychiatric disorders remains largely unknown. An increase in the number of emergency room visits of psychiatric patients on extremely hot or humid days has been documented[9,10]. A nationally representative panel study reported that an increase in mean temperature was associated with an increased odd of anxiety, and an increase in mean humidity was associated with the occurrence of depression and anxiety[11]. A consensus has been established that climate change is the greatest global health threat[6,11], and the Intergovernmental Panel on Climate Change predicts that heatwaves and humid heat stress will increase in intensity and frequency in the coming decades[12]. A meta-analysis from regions worldwide supported positive associations between elevated temperatures and mental health-related mortality and morbidity[13]. However, the mechanisms of neuropsychiatric disorders caused by the humid heat environment are still elusive, and corresponding intervention strategies are lacking.

The gut microbiota is an important partner in environmental change-driven diseases by either buffering or exacerbating negative impacts on host populations[14]. Environmental temperature changes may restructure the gut microbiota of wildlife, and microbial responses are involved in many physiological and pathological processes[15,16]. Our previous studies have shown that mice subjected to long-term exposure to a humid heat environment suffer from ecological disorders of the gut microbiota[17]. The central nervous system (CNS) closely interacts with the gut microbiota through the gut–brain axis in response to signals from the internal and external environments[18,19]. Accumulating evidence suggests that some neurological disorders may be attributed to changes in gut microbial diversity. For example, cafeteria diet-induced downregulation of the gut microbiota, including *Streptococcus*, *Lactobacillus* and *Butyrivibrio*, is thought to be the cause of memory deficits in rats[20]. Gut microbial correction was also reported to be a potential mechanism of antidepressant treatment efficacy in a mouse model[21]. The transplantation of faecal microbiota from a healthy young man alleviated anxiety and depression behaviours in patients in a clinical trial[22]. Due to the close relationship between the gut microbiota and neurological function[23], decoding the characteristics of the gut microbiota under specific environmental conditions contributes to developing therapeutic strategies for neuropsychiatric disorders.

The gut microbiota is considered a metabolic "organ" involved in metabolite production and metabolism regulation in the host[24]. The interplay between the gut microbiota and bile acid metabolism has been widely reported[25,26]. Primary bile acids are synthesised from cholesterol in the liver and further digested into secondary bile acids by enzymes derived from intestinal bacteria. Most secondary bile acids are reabsorbed to enter the serum of the circulation system and serve as key signalling molecules in communication between the gut and the CNS to modulate neuroinflammation and synaptic plasticity in the brain[27].

In this work, we studied the impact of a humid heat environment on the brain and the gut microbiota using a conditioned housing mouse model. Neurological disorders were investigated using behavioural tests, electrophysiological recordings, neurotransmitter expression analysis, neuroinflammation, and screens of altered genes in the cortex. Changes in the gut microbiota and metabolites were studied using 16 S rRNA sequencing and mass spectrometry, respectively, and their contributions to neurological disorders were further evaluated by faecal microbiota transplantation in germ-free (GF) mice and probiotic administration in mice exposed to a humid heat environment. Our work revealed that a humid heat environment impaired the gut microbiota (e.g., *Lactobacillus murinus* downregulation) and bile acid metabolism (e.g., lithocholic acid upregulation) and drove the progression of neuroinflammation, a cortical excitatory/inhibitory (E/I) imbalance and anxiety-like behaviours, and probiotic administration reversed these abnormalities. Similar changes in the gut microbiota and serum lithocholic acid (LCA) level were also detected in the subject population during the humid heat season. These studies revealed the impact of a humid heat environment on anxiety disorders and the underlying mechanisms, which will help us develop intervention strategies.

## Results

### Humid heat environment causes anxiety-like behaviour and disrupts the cortical excitatory/inhibitory (E/I) balance in mice

Adult male mice were housed in either an ordinary environment (the NC group; 22–24 °C and 45–55% humidity) or a humid heat environment (the HHE group; 31–33 °C and 91–95% humidity) for 45 days and subsequently were subjected to a series of behavioural tests (Fig. 1a). The HHE group spent less time crossing the central area in the open field test (Fig. 1b) and spent less time in the open arms of the elevated plus maze (Fig. 1c) than did the NC group, indicating anxiety-like behaviour and high stress in the HHE group. Anxiety and depression are associated with overlapping pathways and occur together in many patients[28]. Depression-like behaviours were further assessed using tail suspension, forced swimming and sucrose preference tests. The immobility times in suspension and swimming tests and the sucrose intake ratios were comparable between the two groups (Fig. 1d–f), indicating no obvious depressive behaviours in the HHE group. However, we did not observe behavioural abnormalities in adult female mice housed in a humid heat environment (Supplementary Fig. 1). Therefore, we used only males in this study.

We recorded spontaneous EPSCs and IPSCs from pyramidal neurons in layers II/III of the mPFC using acute brain slices to decode potential mechanisms of the anxiety-like behaviours. Compared with that in the NC group, the mean frequency of sEPSCs in the HHE group was significantly increased (Fig. 1g, h), but the mean amplitude of sEPSCs did not change (Fig. 1i). The mean frequency and amplitude of sIPSCs were similar between the two groups (Fig. 1j–l). These results suggested that the E/I ratio of pyramidal neurons was increased in the HHE group. The EPSC and IPSC frequencies are highly dependent on presynaptic inputs. We then studied the expression of synaptic vesicle proteins in layers II/III of the mPFC by immunostaining for synaptophysin (presynaptic marker), GAD65 (a marker for inhibitory neurons), vGlut1 (a marker for excitatory vesicles) and vGAT (a marker for inhibitory vesicles). In layers II/III of the mPFC, the distribution and densities of synaptophysin, GAD65, and vGAT immunoreactivity were comparable between the two groups (Supplementary Fig. 2a, b, d). An increase in vGlut1 expression surrounding cortical neurons was readily detected in the HHE group (Supplementary Fig. 2c). Cortical samples, including mPFC samples, were subjected to Western blot analysis to quantify the expression of these proteins. The protein levels of vGlut1, but not those of vGAT, GAD65 or synaptophysin, were significantly increased in the HHE group compared to the NC group (Supplementary Fig. 2e). These findings indicate that pyramidal neurons in the HHE group receive more excitatory inputs (vGlut1+ vesicles) than those in the control group, supporting the increased frequency of sEPSCs in the electrophysiological recordings.

### Humid heat environment alters the gut microbiota composition and related serum metabolites

Clinically, dysfunction of the digestive system is a typical early symptom caused by a humid heat environment. Consistent with these findings, we found that the body weight of the HHE group was significantly lower than that of the NC group, and the caecum size decreased in the HHE group (Fig. 2a, b). Caecum size is thought to correlate with the composition of the intestinal microbiota[29]. We

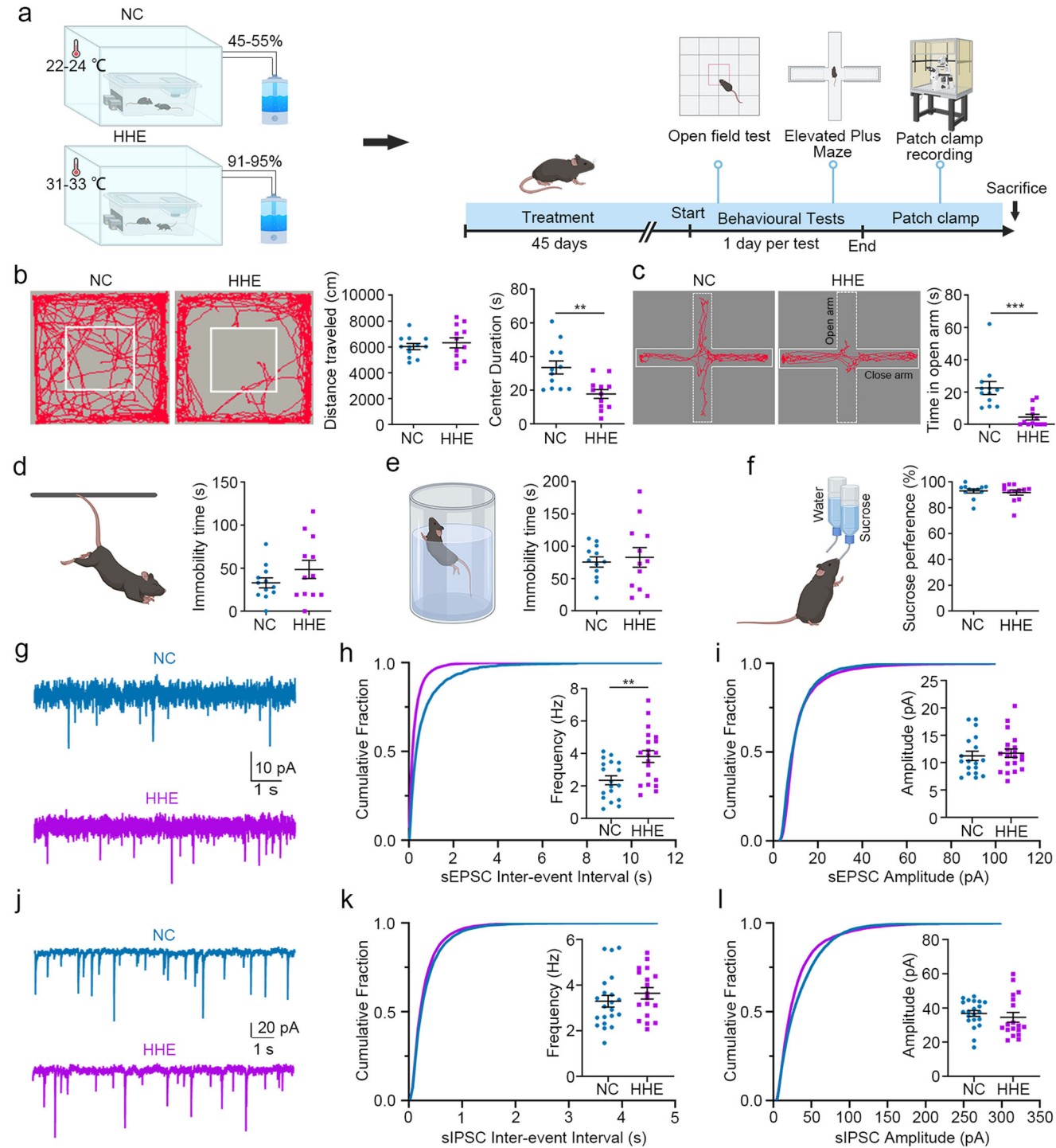

**Fig. 1 | Humid heat environment causes anxiety-like behaviour and disrupts the cortical E/I balance in mice. a** Male mice were randomly divided into the HHE and NC groups (*n* = 12 mice/group), and the experimental design is illustrated in the schematic overview. HHE humid heat environment. NC normal control. Behavioural tests were performed after 45-days conditioned housing. **b** Representative traces and statistical analysis of open-field test. Centre duration, statistical analysis by two-tailed *t*-test, *t*(22) = 3.37, *P* = 0.0028. **c** Representative traces and statistical analysis of elevated plus maze. Time in open arm, statistical analysis by two-tailed *t*-test, *t*(22) = 4.107, *P* = 0.0005. **d**, **e** In two groups, the immobility time was comparable in the tail suspension test and the forced swimming. **f** The percentage of sucrose

consumption was similar in the sucrose preference test. **g**–**i** There was a significant increase of the cumulative sEPSC frequency but no differences of sEPSC amplitude in the HHE group (*n* = 18 neurons in the NC group and 20 neurons in the HHE group). Frequency, statistical analysis by two-tailed *t*-test, *t*(36) = 3.37, *P* = 0.0034. **j**–**l** The sIPSC cumulative frequencies and amplitudes were comparable in two groups (*n* = 21 neurons in the NC group and 18 neurons in the HHE group). Statistical analysis by two-tailed *t*-test. \*\**P* < 0.01; \*\*\**P* < 0.001; *n* = 12 mice/group in b-f and 4 mice/group in **g**–**l**. All data are presented as mean values +/− SEM. Illustrations created with BioRender.com. Source data are provided as a Source data file.

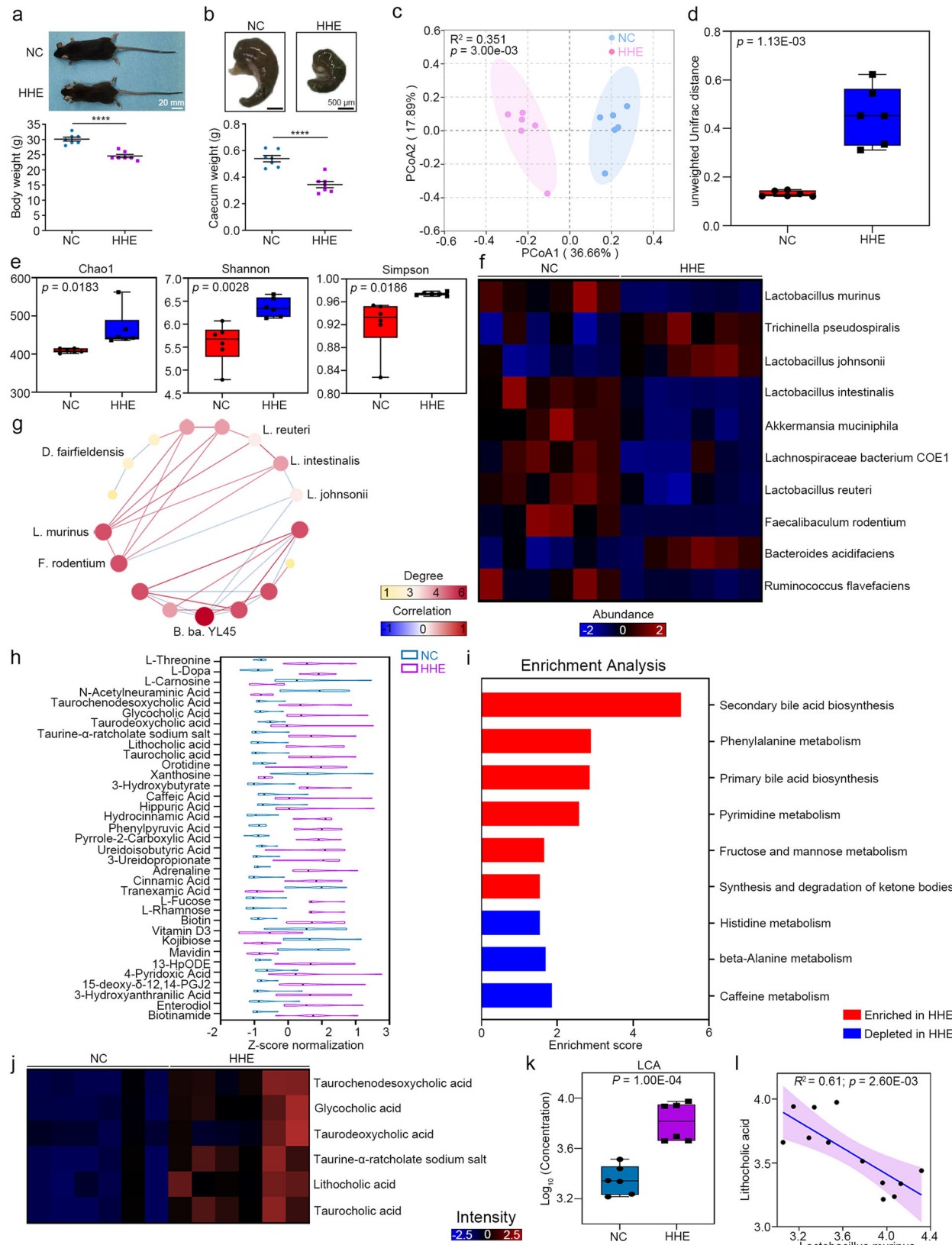

studied the gut microbiota of mouse faecal samples before (baseline) and 45 days after the start of conditioned environment housing using 16 S rRNA gene sequencing. In the two groups, the microbiome composition was similar at baseline in terms of alpha and beta diversity (Supplementary Fig. 3a, b). After 45 days, the beta diversity analysis of the faecal samples revealed that the clustering of the gut microbiota differed markedly between the two groups (Fig. 2c, d), and alpha diversity was greater in the HHE group (Fig. 2e). Compared with those in the NC group, ten bacterial groups in the HHE group exhibited significant changes (Fig. 2f; $P < 0.05$, fold change > 1.5), including a decreased abundance of *L. murinus* and rarely detectable levels of several probiotic bacteria (e.g., *L. intestinalis*, *L. reuteri* and

**Fig. 2 | Humid heat environment impairs gut microbiota and gut microbiota-related metabolites. a**, **b** Body weight. Statistical analysis by two-tailed *t*-test, $t(12) = 6.755$, $P = 0.00002$. $n = 7$ mice/group. Caecum weight. Statistical analysis by two-tailed *t*-test, $t(12) = 5.883$, $P = 0.00007$. $n = 7$ mice/group. ****$P < 0.0001$. **c**, **d** Unweighted UniFrac distance-based analysis (permutational multivariate analysis of variance) and the Boxplots (Tukey's test). $n = 6$ mice/group. **e** Alpha diversity analysis (Chao1, Shannon and Simpson) in two group. Two-tailed Mann–Whitney *U* test. n = 6 mice/group. **f** Total 10 bacteria with different abundance ($P < 0.05$ in false discovery rate (FDR), Foldchange >1.5; blue to red: low to high abundance) were presented by the heatmap in two groups. Two-tailed *t*-test, FDR with two-stage step-up method of Benjamini. **g** Correlation strengths more than 0.6 were shown in the ecological network (**g**; measured by the SparCC method). $n = 6$ mice/group. **h** Violin plot showed the serum metabolites with differential concentration (statistical analysis by two-tailed Mann–Whitney *U* test, normalisation processing by z-score) in two groups. $n = 6$ mice/group. **i** Enrichment analysis disclosed the main metabolism pathways (enrichment scores >1 included). **j** In the secondary bile acid biosynthesis pathway, the expression abundance of 6 metabolites was presented in the heatmap (blue to red colour: low to high abundance; $P < 0.05$, two-tailed Mann–Whitney *U* test). **k** LCA concentration measured by targeted mass spectrometry in the serum ($P < 0.05$, statistical analysis by Two-tailed *t*-test). $n = 6$ mice/group. LCA Lithocholic acid. **l** Linear association analysi of the concentration of LCA and the abundance of *Lactobacillus murinus*, statistical analysis by partial's spearman correlation. $n = 6$ mice/group. In box plot (**d**, **e**, **k**), the lines from top to bottom represent maximum, 3rd quartile, median, 1st quartile, and minimum, while the middle area represents the interquartile range. All data are presented as mean values +/− SEM. Source data are provided as a Source data file.

*Akkermansia muciniphila*) (Fig. 2f). The ecological network interaction analysis revealed co-occurrence correlations between *L. murinus* and two probiotic bacteria (*L. intestinalis* and *L. reuteri*) among the differentially abundant bacteria in the HHE group (Fig. 2g), suggesting synergistic associations between the decreased abundance of *L. murinus* and decreased levels of protective bacteria.

Dysbiosis of the gut microbiota can lead to disturbances in serum metabolites[30,31]. We further analysed the serum metabolites of the gut microbiota using liquid chromatography–mass spectrometry (MS)/MS. Orthogonal partial least squares discriminant analysis revealed that the serum metabolic profile in the HHE group differed significantly from that in the NC group (Supplementary Fig. 4a), including thirty-five metabolites with altered expression (adjusted $P < 0.05$; Fig. 2h, Supplementary Data 1). These metabolites were clustered in different metabolomic pathways, such as secondary bile acid biosynthesis, phenylalanine metabolism, primary bile acid biosynthesis, primary metabolism, and fructose and mannose metabolism (Fig. 2i). Among them, secondary bile acid biosynthesis was the first ranked pathway, and an increase in the levels of bile acid, including taurochenodeoxycholic acid, glycocholic acid, taurodeoxycholic acid, taurine-α-ratcholate sodium salt, lithocholic acid (LCA) and taurocholic acid, was observed in the HHE group compared to the NC group (Fig. 2j). Compared with those in the NC group, targeted mass spectrometry of mouse serum and brain samples revealed the significant upregulation of LCA in the HHE group (Fig. 2k, Supplementary Fig. 5).

Gut bacteria maintain bile salt hydrolase activity and influence intestinal bile acid deconjugation and excretion[32]. In the two groups, the levels of gut bacterial phyla were measured using RT–PCR, and the results revealed a significant decrease in the abundance of Firmicutes and a significant increase in the abundance of Bacteroidetes in the HHE group (Supplementary Fig. 4b), in which bile salt hydrolase activity is high in Firmicutes and low in Bacteroidetes[33]. The potential correlations between the intestinal microbiota and serum metabolites were analysed using partial Spearman's correlation coefficients. In the HHE group, low abundances of *L. murinus*, *L. intestinalis* and *L. reuteri* were negatively correlated with LCA expression (Supplementary Fig. 6). Similarly, the abundance of *L. murinus* was negatively correlated with the serum LCA concentration (Fig. 2l). Thus, gut microbial dysbiosis leads to alterations of serum metabolites in the HHE group.

Exploiting the seasonal changes in environmental humidity and temperature and to be consistent with animal studies, serum and faecal samples were collected from male human subjects only during two periods with obvious differences (phase 1: 22–25 °C and 30–60% humidity; phase 2: 32–35 °C and 90–95% humidity), and the duration of the relatively stable microenvironment in each phase was approximately 4 weeks (Supplementary Fig. 7a). Notably, 16 S rRNA gene sequencing of the faecal samples revealed significant changes in the structure of the human gut microbiota during phases 1 and 2, mainly in terms of marked differences in beta diversity and significant increases in α diversity (observed species and Chao1) (Supplementary Fig. 7b, c). In these subjects, the abundance of *L. murinus* in the faecal samples decreased, and the serum LCA levels increased in phase 2 compared to phase 1 (Supplementary Fig. 7d, e). Additionally, serum TNF-α levels tended to increase in phase 2 compared to phase 1 (Supplementary Fig. 7f).

## Transplantation of the faecal microbiota from the HHE group recapitulates alterations in the gut microbiota and anxiety-like behaviours in GF mice

GF mice received a faecal microbiota transplant through the oral gavage of mouse stools from either the HHE group or the NC group (referred to as the GF-HHE and the GF-NC groups, respectively) to confirm the direct role of the altered gut microbiota induced by the humid heat environment. Faecal samples were harvested from the GF-HHE and GF-NC groups 14 days after stool gavage, followed by 16 S rRNA sequencing (Fig. 3a). PCoA revealed that the gut microbiota diversity of the GF recipient mice was similar to that of their corresponding donor mice (Supplementary Fig. 8). Compared to that in the GF-NC group, the alpha diversity of the gut microbiota in the GF-HHE group was significantly greater, similar to that in the HHE group (Fig. 3b). The beta diversity of the gut microbiota was distinctly segregated between the two groups (GF-HHE vs. GF-NC, $P < 0.01$, PERMANOVA; Fig. 3c). Among the 14 bacteria with altered abundance (Fig. 3d), *L. murinus*, *L. intestinalis*, *L. reuteri* and *Akkermansia muciniphila* were significantly less abundant in the GF-HHE group than in the GF-NC group ($P < 0.05$, fold change >1.5; Fig. 3d). Comparisons of the GF-HHE group with the GF-NC group (GF mice) and the HHE group with the NC group (conventional mice) revealed similar changes in bacterial abundance (Fig. 3e), indicating successful colonisation. Decreased abundances of *L. murinus* and the probiotics *L. intestinalis*, *L. reuteri* and *Akkermansia muciniphila* were detected in both the HHE group and the GF-HHE group (Fig. 3e). The correlation analysis revealed that the ecological network modules were conserved after faecal transplantation in the GF-HHE group compared to those in the HHE group (Fig. 3f). Moreover, the serum LCA level was also significantly increased in the GF-HHE group compared to the GF-NC group (Fig. 3g).

Two weeks after faecal microbiota transplantation, the GF mice underwent behavioural tests. The mice in the GF-HHE group spent less time travelling through the central area in the open-field test (Fig. 3h, i) and spent less time in the open arms of the elevated plus maze (Fig. 3j, k) than did the GF-NC group, indicating that microbiota transplantation reproduced the anxiety-like behaviours of the HHE group in the GF mice. Acute brain slices, including those from the mPFC, were prepared for electrophysiological recordings 2 weeks after transplantation. In the GF-HHE group, the mean frequency of sEPSCs in pyramidal neurons was significantly greater than that in the GF-NC group (Fig. 3l, m). The mean amplitude of sEPSCs was comparable between the two groups (Fig. 3n). The mean frequency and amplitude of sIPSCs were indistinguishable between the GF-HHE group and the GF-NC group (Fig. 3o, p, q). Electrophysiological studies indicate an increase in excitability in pyramidal neurons after microbiota

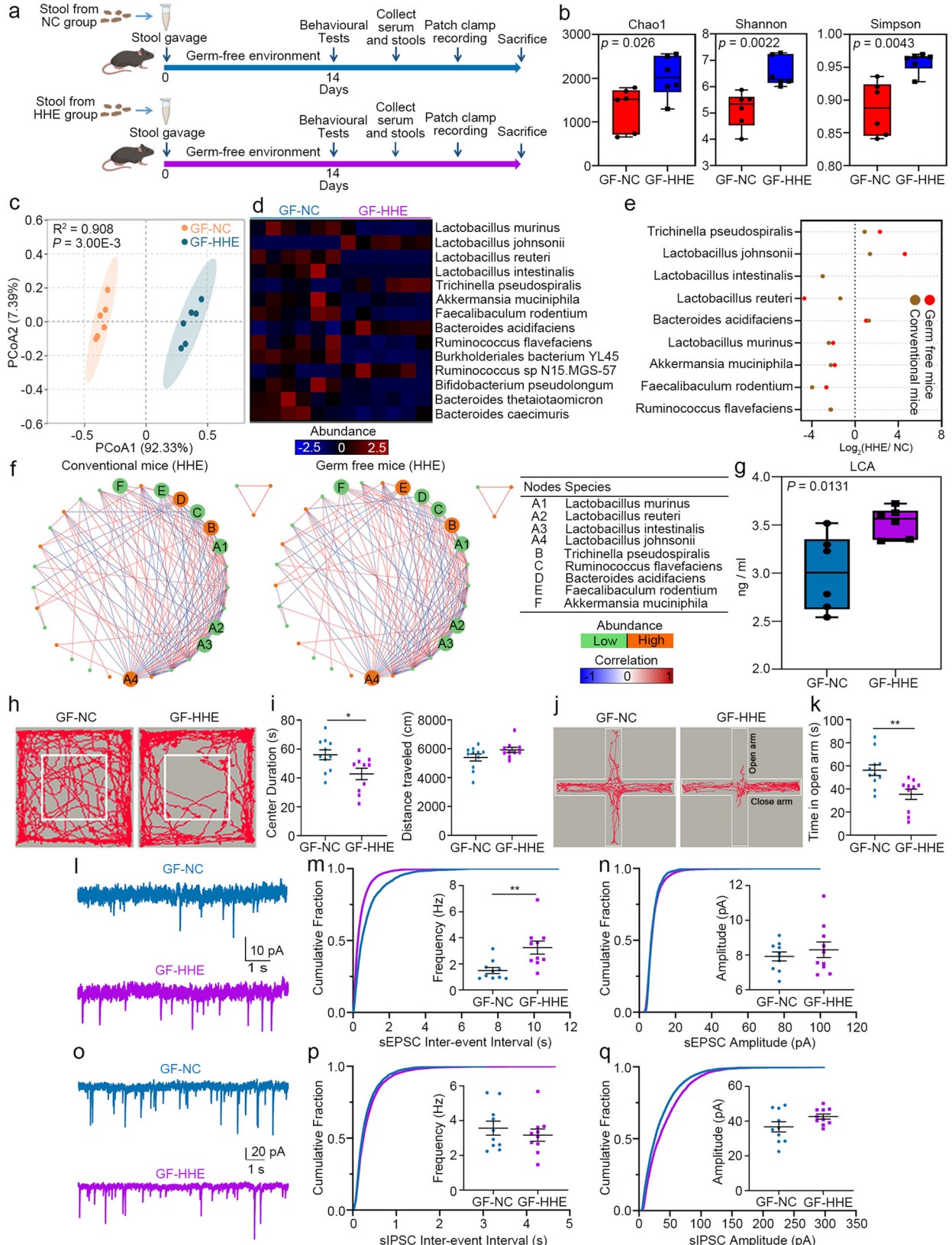

transplantation. In support of this finding, levels of the vGlut1 protein were quantified in cortical samples using Western blotting, which revealed a significant increase in the GF-HHE group compared with the GF-NC group (Supplementary Fig. 9). Moreover, bacteria in the intestinal tract were depleted by the administration of an antibiotic cocktail (ampicillin, 0.25 mg/mL; neomycin, 0.25 mg/mL;

metronidazole, 0.25 mg/mL; vancomycin, 0.125 mg/mL), and the gut microbiota was subsequently reconstructed by faecal microbial transplantation (FMT) from the NC and HHE groups of mice to exclude the possibility that the gut–brain axis was markedly altered in germ-free animals in this study. We found that FMT recapitulated the altered molecular features and anxiety-like behaviours in FMT mice

**Fig. 3 | Faecal transplantation recapitulates gut microbiota alteration and anxiety-like behaviours in GF mice. a** The schematic overview illustrated the experiment design, in which GF mice were orally gavaged with stool from mice in the NC group or the HHE group, defined as the GF-NC group or GF-HHE group respectively. **b** 16S-RNA sequencing of faecal samples and alpha diversity of Chao1, Shannon and Simpson analysis (two-tailed Mann–Whitney $U$ test). $n = 6$ mice/ group. **c** Beta diversity of PCoA ratio (PERMANOVA). **d** Heatmap showed total 10 bacteria with differential abundance ($P < 0.05$ in FDR, Foldchange >1.5). Two-tailed $t$-test, FDR with two-stage step-up method of Benjamini. **e** The abundance alteration of 10 gut bacteria was calculated in the conventional mice (the HHE group versus the NC group) and the GF mice (the GF-HHE group *versus* the GF-NC group), the change folds denoted as Log$_2$ (HHE/NC) **f** Network modules of altered gut bacteria (measured by the SparCC method). **g** Serum LCA concentration. Statistical

analysis by two-tailed $t$-test. $n = 6$ mice/group. **h–k** Representative traces and statistical analysis of open-field test and elevated plus maze. Centre duration, statistical analysis by two-tailed $t$-test, $t(19) = 2.566$, $P = 0.0189$. Time in open arm, statistical analysis by two-tailed $t$-test, $t(19) = 3.227$, $P = 0.0044$. GF-NC: $n = 11$ mice, GF-HHE: $n = 11$ mice. **l–n** Electrophysiological recordings of sEPSC frequency and the amplitude. Frequency, statistical analysis by two-tailed $t$-test, $t(18) = 3.186$, $P = 0.0051$, 10 neurons from 3 mice in each group). **o–q** sIPSC frequency and amplitude (statistical analysis by two-tailed $t$-test, 10 neurons from 3 mice in each group). In box plot (**b**, **g**), the lines from top to bottom represent maximum, 3rd quartile, median, 1st quartile, and minimum, while the middle area represents the interquartile range. All data are presented as mean values +/− SEM. *$P < 0.05$; **$P < 0.01$. Illustrations created with BioRender.com. Source data are provided as a Source data file.

(Supplementary Fig. 10). Taken together, these findings indicate that the presence of a mouse faecal microbiota from HHE mice is sufficient to drive the E/I imbalance of pyramidal neurons in the mPFC and the onset of anxiety-like behaviour, indicating that the alteration of the gut microbiome induced by a humid heat environment is the direct cause of anxiety-like behaviour.

## The HHE and GF-HHE groups exhibit increased permeability of the gut barrier and BBB and an enhanced inflammatory response

Alterations in the gut microbial composition lead to increased permeability of the gastrointestinal tract and BBB impairment, inducing neuroinflammatory processes involved in neurological disorders, such as anxiety and depressive-like disorders[34,35]. We first analysed the expression levels of the tight junction proteins claudin-1 and ZO-1to investigate the impact of HHE on gut barrier function and the important role of the intestinal microbiota. In the HHE and GF-HHE groups, the expression of claudin-1 and ZO-1 was significantly decreased, as shown by the Western blot analysis (Fig. 4a, Supplementary Fig. 11a) and immunofluorescence staining (Fig. 4b, Supplementary Fig. 11b). Two hours after FITC-dextran gavage, significantly higher serum FD4 levels were detected in the HHE and GF-HHE groups than in the NC and GF-NC groups (Fig. 4c, Supplementary Fig. 11c), indicating increased gut permeability. Inflammatory factors including G-CSF, GM-CSF, IFN-γ, IL-1b, IL-6, IL-9, IL-17A, KC and TNF-α, were significantly upregulated in the HHE group compared to the NC group (Fig. 4d). The levels of inflammatory factors, including IFN-γ and TNF-α, were also significantly increased in serum samples from the GF-HHE group (Supplementary Fig. 11d). The intestinal structure was obviously atrophied with a reduced villus height in the HHE and GF-HHE groups (Fig. 4e, Supplementary Fig. 11E). BBB permeability was studied by Evans blue tracing (Fig. 4f, Supplementary Fig. 11f). After tail vein delivery, infiltrated Evans blue dye was readily detected in the brains of the HHE and GF-HHE groups but was rarely detected in the NC and GF-NC groups (Fig. 4g–k, Supplementary Fig. 11g–k). Moreover, the expression of claudin-1, occludin-1 and ZO-1 was significantly downregulated in the HHE and GF-HHE groups compared to the NC and GF-NC groups, as shown by Western blot analysis (Fig. 4l, Supplementary Fig. 11l).

As described previously, increased BBB permeability and serum inflammatory factor levels were detected in the HHE and GF-HHE groups. We further examined the expression of inflammatory factors in the brain using Bio-Plex and ELISA. As expected, the levels of the proinflammatory cytokines IFN-γ and TNF-α were significantly increased in the HHE group compared to the NC group (Fig. 4m). Furthermore, after GF mice received the gut microbiota transplant (Supplementary Fig. 11m), the expression of IFN-γ and TNF-α in cortical samples from the GF-HHE group was upregulated, as detected using ELISA group (Supplementary Fig. 11o). In the brain, microglial activation and microglia–neuron interactions have important effects on neuronal morphology and function. In the cortical sections immunostained for Iba1 (microglial marker) and NeuN (neuronal marker),

the interactions of microglia and neurons were classified as follows: microglial soma in close contact with neuronal soma with their processes either further enwrapping the neuron (category I) or not (category II), only microglial processes in contact with neuronal soma (category III), and microglia with no close contact with neurons (category IV). The ratios of different categories of microglia were subsequently calculated (Fig. 4n, Supplementary Fig. 11n). In the HHE and GF-HHE groups, the proportion of category I microglia was significantly greater than that in the NC and GF-NC groups (Fig. 4n, Supplementary Fig. 11n), indicating an increase in microglia–neuron interactions. Active microglia are the origin of macrophages (positive for CD68) in the brain and exhibit morphological changes, such as an enlarged soma size and shortened processes. In the immunostained cortical sections, significant increases in the microglial density, number of Iba1 + CD68+ cells, and size of the microglial soma and a significant decrease in the number of microglial branches were observed in the HHE and GF-HHE groups compared to those in the NC and GF-NC groups (Fig. 4o, Supplementary Fig. 11p).

Moreover, we further studied the key role of LCA (which is strongly correlated with *L. murinus*) in inflammation and the BBB, as mentioned above. We performed a tail vein injection of LCA for 7 consecutive days (Supplementary Fig. 12a), and then analysed the expression levels of inflammatory factors and tight junction proteins (ZO-1, claudin-1 and occludin-1) in the brain. Compared with the saline injection, LCA treatment significantly increased the expression of TNF-α and IL-6 (Supplementary Fig. 12b) and downregulated the expression of tight junction proteins in the LCA injection group (Supplementary Fig. 12c). Elevated LCA levels cause liver damage, and thus we performed H&E staining of liver sections and detected the expression of TNF-α and IL-6 in liver samples from the HHE and NC groups. Our results showed that lipid deposition (red arrows indicate areas of vacuoles) and the levels of proinflammatory cytokines (TNF-α and IL-6) were increased in the HHE group compared with the NC group (Supplementary Fig. 13a, b). Taken together, these findings indicate that a humid heat environment impairs the gut and BBB and upregulates serum inflammatory factors, which further influence the inflammatory state in the brain, and that alterations in the gut microbiota and abnormal metabolism may play important direct roles.

## PI3K/AKT/NF-κB signalling is enhanced in the brains of mice from the HHE and GF-HHE groups

We performed RNA sequencing of cortical samples after 45 days of conditioned housing to identify transcriptomic changes in the cortex. Compared to those in the NC group, 391 upregulated genes and 48 downregulated genes were identified in the HHE group (Fig. 5a). KEGG analysis revealed that DEGs clustered in many inflammation-related signalling pathways: the PI3K-AKT pathway ranked first, followed by other pathways, such as NF-kB signalling and TNF signalling (Fig. 5b). GSEA revealed a significant upregulation of PI3K-AKT signalling in the HHE group compared to the NC group (Fig. 5c). In this pathway, the increases in the PI3K, AKT1 and NF-κB1 mRNAs were further confirmed

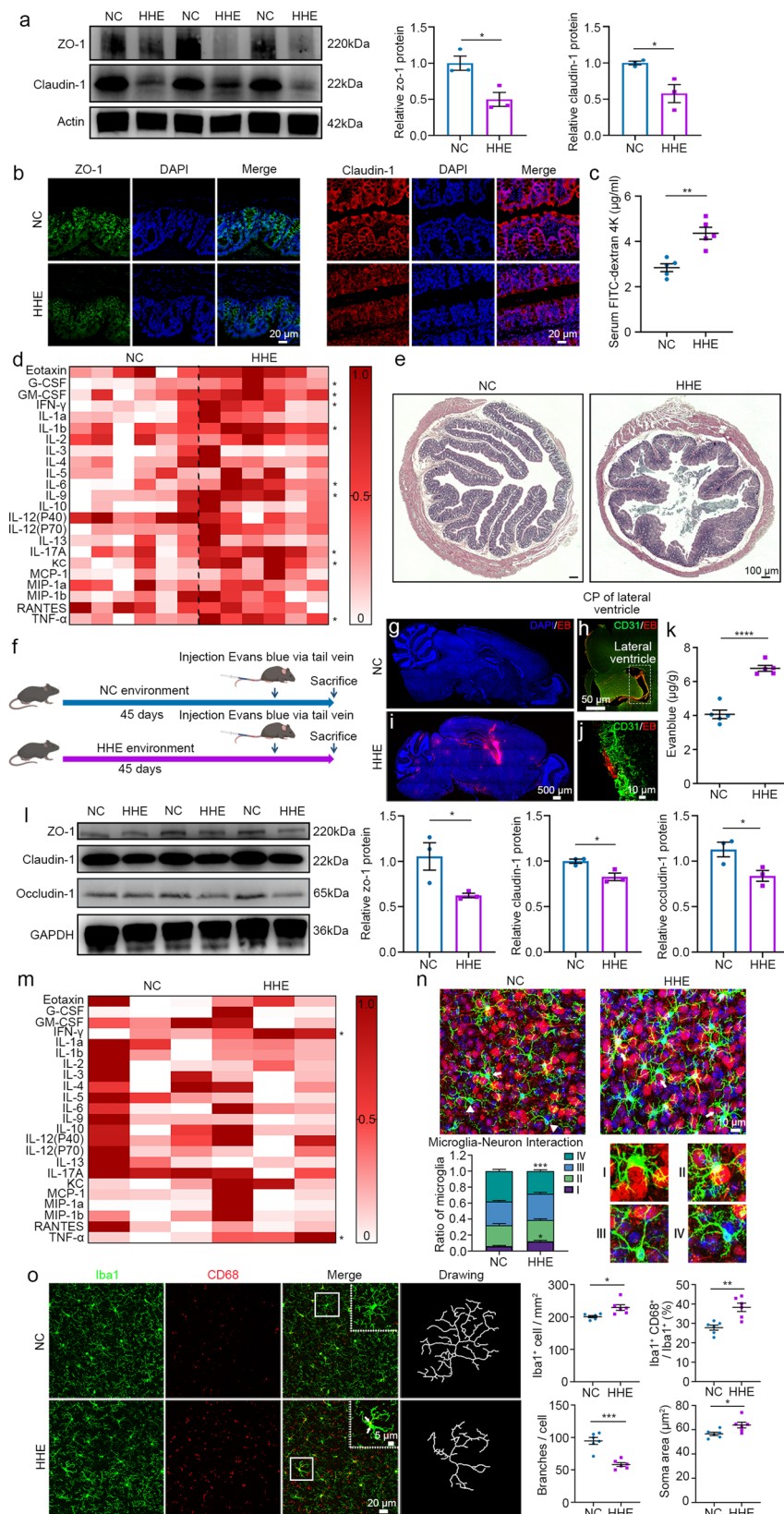

in the HHE group via RT–PCR (Fig. 5d). Western blots of cortical samples from the HHE group showed significantly increased levels of the phosphorylated PI3K and AKT proteins, as well as the NF-κB protein, compared to the NC group (Fig. 5e, f). Furthermore, a positive correlation was observed between the phosphorylation of PI3K and AKT and the serum LCA level (Fig. 5g). Similar changes were identified

in the GF mice after faecal transplantation: the levels of phosphorylated PI3K and AKT proteins and NF-κB protein were increased in the GF-HHE group compared to those in the GF-NC group (Fig. 5h, i) and LCA levels were positively correlated with the phosphorylation of PI3K and AKT (Fig. 5j). In addition, the mRNA levels of C-X-C motif chemokine ligand 1 (Cxcl1) and C-X-C motif chemokine receptor 2 (Cxcr2),

**Fig. 4 | Humid heat environment causes the impairment of gut barrier and BBB and activates neuroinflammation. a, b** Western blots and immunofluorescence staining of the colon sections. DAPI (blue) counterstained nuclei. Statistical analysis by two-tailed *t*-test. Relative zo-1 protein, $t(4) = 3.647$, $P = 0.0218$. Relative claudin-1 protein, $t(4) = 3.302$, $P = 0.0299$. $n = 3$ mice/group. **c** Serum FITC-dextran. Statistical analysis by two-tailed *t*-test, $t(8) = 4.785$, $P = 0.0014$. $n = 5$ mice/group. **d** Heatmap showed the expression of inflammatory cytokines in the serum. Statistical analysis by multiple *t*-test with the Holm-Sidak method, $n = 6$ mice/group. **e** H.E. stain of colon tissue. $n = 3$ mice/group. **f** Schematic diagram showing the experimental procedures. **g, i** Abundant EB signal (red) was visible in the HHE group, but rare in the NC group. **h, j** EB-positive signal accumulated surrounding blood vessels. **k** The concentration of EB in the brain. Statistical analysis by two-tailed *t*-test, $t(8) = 11.29$, $P = 0.00002$. $n = 5$ mice/group. **l** Representative image and statistical analysis of the claudin-1, occludin-1 and zo-1 protein levels. Statistical analysis by two-tailed *t*-test. Relative zo-1 protein, $t(4) = 2.816$, $P = 0.048$. Relative occludin-1 protein, $t(4) = 2.893$, $P = 0.0444$. Relative claudin-1 protein, $t(4) = 3.593$ $P = 0.0229$. $n = 3$ mice/group. **m** Heatmap showed the expression of inflammatory cytokines in the cortical samples. Statistical analysis by multiple *t*-test with the Holm-Sidak method, $n = 3$ mice/group. **n** In cortical sections immunostained for Iba1 (green) and NeuN (red), white arrows point to I type and white triangles point to IV type. Statistical analysis by two-way ANOVA with Bonferroni's post hoc test. Ratio of microglia (I type), $F(3, 40) = 70.56$, $P = 0.0338$. Ratio of microglia (IV type), $F(3, 40) = 70.56$, $P = 0.0009$. $n = 6$ mice/group. **o** Representative images of microglia and statistics showed Iba1-positive cell density ($t(10) = 2.996$, $P = 0.0134$), the percentage of double positive cells ($t(10) = 4.136$, $P = 0.002$), average microglial soma area ($t(10) = 2.644$, $P = 0.0246$) and average microglia branch number ($t(10) = 5.998$, $P = 0.0001$). Statistical analysis by two-tailed *t*-test. $n = 6$ mice/group. All data are presented as mean values +/− SEM. *$P < 0.05$; **$P < 0.01$; ***$P < 0.001$; ****$P < 0.0001$. Illustrations created with BioRender.com. Source data are provided as a Source data file.

two key molecules in the TNF-inflammatory pathway, were significantly increased in the GF-HHE group compared to the GF-NC group, as shown by RT–PCR (Fig. 5k). These findings indicate that the microbiota alterations induced by the humid heat environment activate PI3K/AKT/NF-κB signalling and neuroinflammation in the cortex.

### *L. murinus* administration reverses abnormalities in mice from the HHE group

As described above, for *L. murinus*, the abundances of two other protective bacteria, *L. reuteri*[36] and *Akkermansia muciniphila*[37], which have been reported to improve anxiety-like behaviour, decreased. Therefore, we tested whether *L. murinus*, *L. reuteri* and *Akkermansia muciniphila* improve anxiety-like behaviour induced by humid heat environments. HHE-exposed mice were administered *L. murinus* (the HHE + L group), *L. reuteri* (the HHE + L. reuteri group), *Akkermansia muciniphila* (the HHE+Akk group), or saline (the HHE + S group). In the open-field test, the time that mice in the HHE + L and HHE + L. reuteri groups spent in the central area increased, but not for mice in the HHE +Akk group, compared with the HHE + S group (Supplementary Fig. 14a). In the elevated plus maze test, the time spent in the open arms was increased in the HHE + L group, HHE + L. reuteri group and HHE+Akk group (Supplementary Fig. 14b). However, *L. murinus* treatment induced the most significant improvements in behaviour. We then asked whether an additional supply of *L. murinus* could reverse these phenotypes. Mice in the HHE group were orally gavaged with *L. murinus* and the same volume of saline as a control to test this hypothesis (Fig. 6a). After 14 days of oral gavage, we collected faecal samples from the HHE + L and HHE + S groups and performed 16 S rRNA gene sequencing. In the HHE + L group (treated with *L. murinus*), the beta diversity of the gut microbiota differed significantly from that in the HHE + S group (treated with saline) (Supplementary Fig. 15a). The abundances of *L. murinus* and *L. reuteri*, but not *Akkermansia muciniphila*, were significantly greater in the HHE + L group than in the NC group (Supplementary Fig. 15b). We also found that, compared with those in the HHE + S group, the abundances of the phyla Firmicutes and Actinobacteria, which have high bile salt hydrolase activity, were restored in the gut microbiota (Fig. 6b). In the HHE + L group, LCA levels were significantly decreased in the serum but increased in the faecal samples compared to those in the HHE + S group (Fig. 6c, d), suggesting that *L. murinus* administration promotes LCA excretion.

Compared with those in HHE + S mice, anxiety-like behaviours in *L. murinus*-treated HHE mice were efficiently alleviated, as indicated by increased time spent in the central area of the open field and in the open arms of the elevated plus maze (Fig. 6e), and TNF-α expression was decreased in the serum (Fig. 6f). In these animals, we also performed electrophysiological recordings of pyramidal neurons in the mPFC (Fig. 6g, j). In the HHE + L group, the sEPSC frequency was significantly decreased compared to the HHE + S group and comparable to that in the NC group (Fig. 6h), whereas the sEPSC amplitude and

sEPSCs were similar among the different groups (Fig. 6i, k, l). These results indicate that *L. murinus* administration reverses the E/I imbalance in the HHE group.

Furthermore, Western blot analysis of cortical samples revealed that the levels of the phosphorylated PI3K and AKT proteins, as well as the NF-κB protein, were significantly lower in the HHE + L group than in the HHE + S group and were comparable to those in the NC group (Fig. 6m). Thus, *L. murinus* administration also efficiently reversed PI3K/AKT/NF-κB signalling activation in the HHE group.

## Discussion
In this study, we investigated the impact of long-term exposure to a humid heat environment on neuropsychiatric health and the underlying mechanisms using a conditioned housing mouse model. Our main findings included the following: (i) mice housed in a long-term humid heat environment exhibited neuroinflammation in the brain, an E/I imbalance in pyramidal neurons in the mPFC, and anxiety-like neurological dysfunction; (ii) abnormalities in gut microbial abundance and related metabolites induced by the humid heat environment drove the progression of neuropathological changes and neural dysfunction; and (iii) probiotic administration alleviated pathological changes in the brain and neurological deficits by modulating the balance of the gut microbiota and related metabolites (Supplementary Fig. 16). This study provides strong evidence that a humid heat environment harms mental health by impairing the gut microbiota and related metabolites and that regulating the gut microbiota is a potential intervention strategy.

After 45 days of housing in a humid heat environment, the mice preferred to travel in the peripheral region of the open field and stay in the closed arms of the elevated plus maze, which are typical anxiety-like behaviours[38]. These behavioural changes were supported by the increase in the synaptic E/I ratio observed in electrophysiological recordings of pyramidal neurons in the mPFC. The synaptic E/I balance is critical for maintaining brain health, and an increase in the E/I ratio results in excitotoxicity[39], which can lead to anxiety-like behaviours[40]. The excitability of pyramidal neurons depends on synaptic inputs, and the number of excitatory synapses directly influences the frequency of sEPSCs in pyramidal neurons[41]. In our mouse model, the number of excitatory vesicles (vGlut1-positive) was significantly increased in the cortex, as shown by immunofluorescence staining and Western blotting. Therefore, our study confirmed that long-term exposure to humid heat environments is an important risk factor for anxiety disorders, providing a theoretical basis for previous epidemiological investigations[11].

Neuroinflammation is an important cause of the progression of psychiatric disorders, and reactive microglia and proinflammatory cytokines may promote presynaptic glutamate release, thus leading to an E/I balance towards the excitatory state[42]. In the HHE group, neuroinflammation was enhanced in the brain, which was supported by

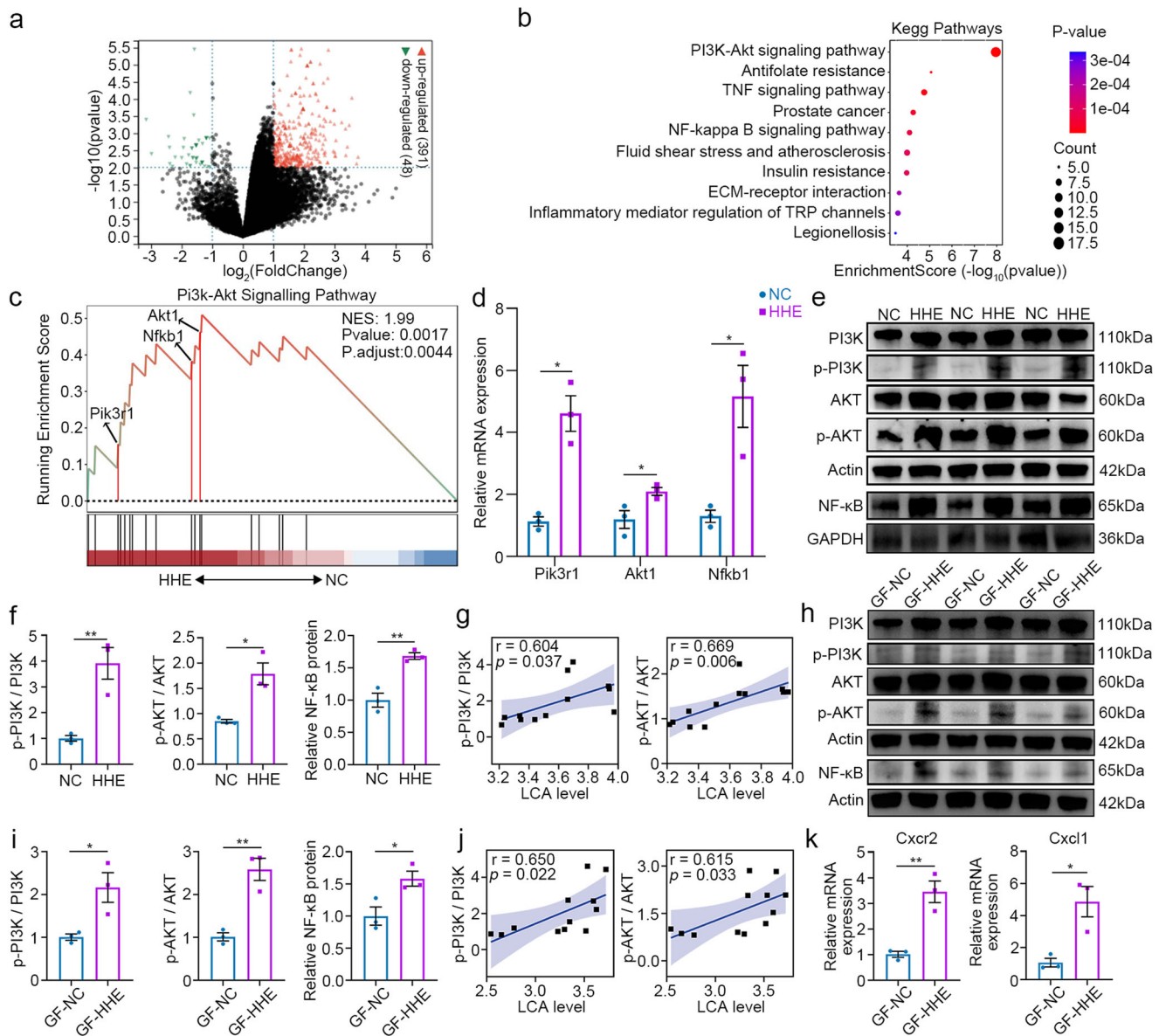

**Fig. 5 | PI3K/AKT/NF-κB signalling is enhanced in the cortical samples from the HHE and GF-HHE groups. a** RNA sequencing identified DEGs between the HHE and the NC groups as shown by the volcano plot including 391 DEGs with upregulation and 48 DEGs with downregulation. **b** Top 10 pathways were clustered using KEGG analysis. **c** GSEA analysis disclosed the upregulation of the PI3K-AKT signalling in the HHE group compared to the NC group. **d** The upregulation of PI3K, AKT1 and NFκB1 was confirmed by RT–PCR. Statistical analysis by multiple *t*-test with the Holm-Sidak method. Relative mRNA expression of Pik3r1, $t(4) = 5.879$, $P = 0.0125$. Relative mRNA expression of Akt1, $t(4) = 2.902$, $P = 0.044$. Relative mRNA expression of Nfkb1, $t(4) = 3.804$, $P = 0.0377$. $n = 3$ mice/group. **e, f** Western blots of cortical samples showed the upregulation in phosphorylated PI3K and AKT proteins and total NF-κB proteins. Statistical analysis by two-tailed *t*-test. p-PI3K/PI3K, $t(4) = 4.66$, $P = 0.0096$. p-AKT/AKT $t(4) = 3.331$, $P = 0.0291$. Relative NF-κB protein,

$t(4) = 5.734$, $P = 0.0046$. $n = 3$ mice/group. **g** The increase of phosphorylated PI3K and AKT was positively correlated to the upregulation of serum LCA using Pearson correlation analysis; $n = 6$ mice/group. **h–j** PI3K, AKT and NF-κB were identified in the GF-HHE group compared to the GF-NC group in western blots. Statistical analysis by two-tailed *t*-test. p-PI3K/PI3K, $t(4) = 3.275$, $P = 0.0307$. p-AKT/AKT, $t(4) = 5.722$, $P = 0.0046$. Relative NF-κB protein, $t(4) = 3.156$, $P = 0.0343$. $n = 3$ mice/group. **k** In the cortical samples, Cxcr2 and Cxcl1 mRNA levels were increased in the GF-HHE group compared to the GF-NC group by RT-PCR. Statistical analysis by two-tailed *t*-test. Relative mRNA expression of cxcr2, $t(4) = 5.611$, $P = 0.005$. Relative mRNA expression of cxcl1, $t(4) = 3.874$, $P = 0.0179$. $n = 3$ mice/group. All data are presented as mean values +/− SEM. *$P < 0.05$; **$P < 0.01$. Source data are provided as a Source data file.

the results described below. First, BBB permeability was increased, as shown by the decreased expression of connexin proteins (Zo-1, Claudin-1 and Occludin-1) and by Evans blue staining. Second, microglia in the brain were highly activated, as indicated by increases in the total density and the proportion of CD68-positive microglia, morphological changes to a reactive state, and remarkably strengthened interconnections between glia and cortical neurons (the percentage of type I microglia increased). Third, the expression of proinflammatory cytokines, including TNF-α and IFN-γ was increased in cortical samples.

Our results revealed that gut microbiota dysbiosis induced by a humid heat environment drives the progression of neurological dysfunction by promoting neuroinflammation in the brain. After long-term exposure to a humid heat environment, gut barrier permeability increased with the downregulation of connexin proteins (Zo-1 and Claudin-1), as determined by FITC tracing, and abundant proinflammatory cytokines (e.g., TNF-α and IFN-γ) were upregulated in mouse serum, indicating a global inflammatory response. Importantly, the gut microbiota composition showed a marked

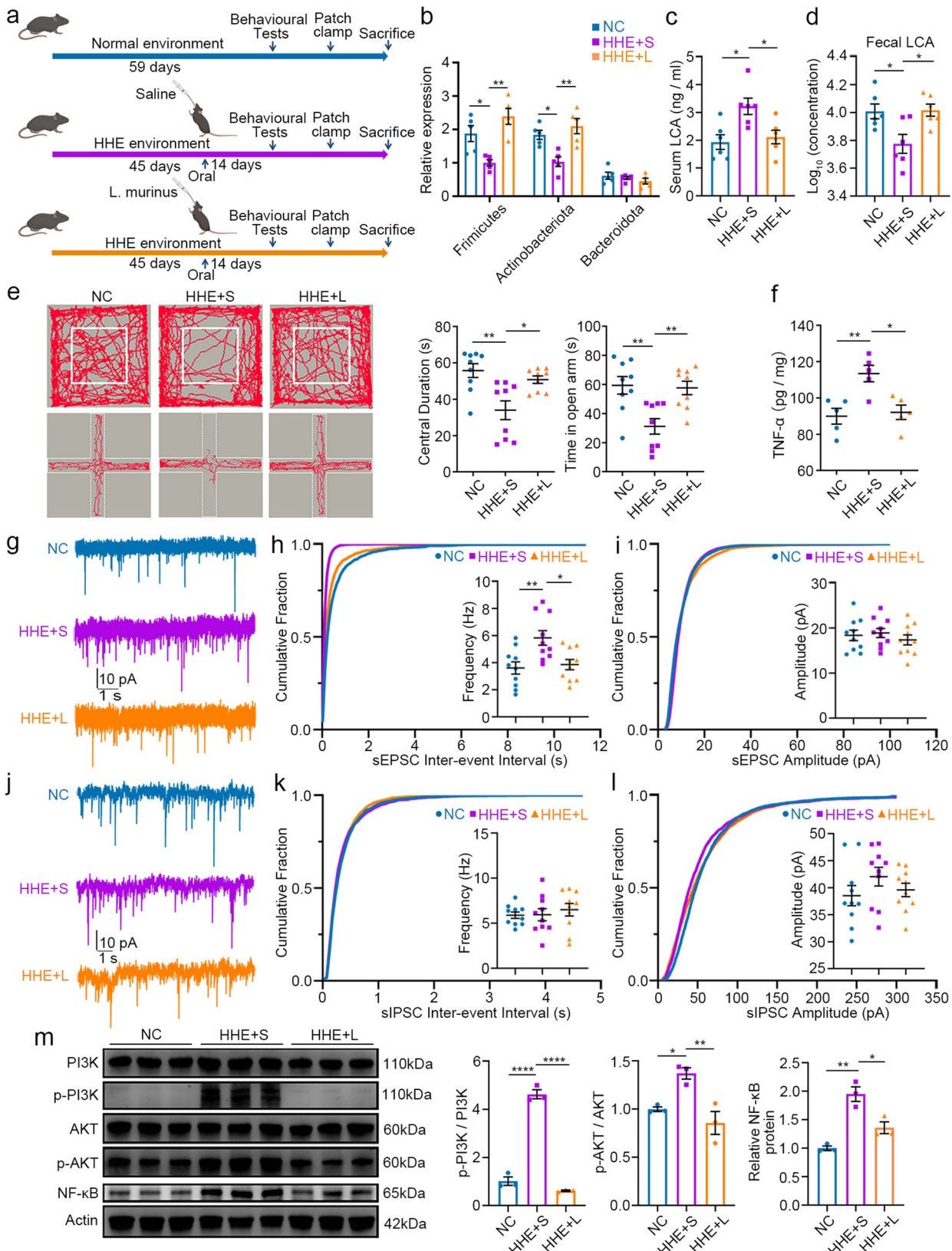

change, and the abundances of many gut microbiota (e.g., *L. murinus*, Akkermansia muciniphila, and *L. reuteri*) were significantly decreased in the HHE group. As a commensal gut bacterium in mammals, *L. murinus* is considered a potential probiotic due to its anti-inflammatory and antibacterial effects[43,44]. Similarly, *Akkermansia muciniphila* and *L. reuteri* have been reported to maintain neuroprotective effects by alleviating neuroinflammation[36,45]. GF mice that received transplantation of faecal samples from humid heat environment-exposed mice fully recapitulated the phenotypes observed in the donor mice. These results suggest that impaired gut microbiota are the initial cause of neuroinflammation and anxiety-like behaviours.

**Fig. 6 | *Lactobacillus murinus* treatment reverses the abnormalities in the HHE group. a** Schematic overview illustrates the experimental design and mice were divided into 3 groups. The NC group: no treatment; the HHE + S group: 14-days oral gavage of physiological saline; the HHE + L group: 14-days oral gavage of live *Lactobacillus murinus*. $n = 12$ mice/group. **b** RT-PCR. Statistical analysis by one-way ANOVA with Bonferroni's post hoc test. The relative expression of Frimicutes, $F(2, 12) = 12.08$, $P = 0.0013$. The relative expression of Actinobacteriota, $F(2, 12) = 10.21$, $P = 0.0026$. $n = 5$ mice/group. **c, d** The LCA levels in the serum and the stool were measured by targeted mass spectrometry assay. Statistical analysis by one-way ANOVA with Bonferroni's post hoc test. Serum LCA, $F(2, 15) = 6.820$, $P = 0.0078$. $n = 5$ mice/group. $Log_{10}$ (concentration), $F(2, 15) = 6.081$, $P = 0.0116$. **e** Representative traces and statistical analysis of open-field test and elevated plus maze. Statistical analysis by one-way ANOVA with Bonferroni's post hoc test. Centre duration, $F(2, 24) = 8.665$, $P = 0.0015$. Time in open arm, $F(2, 24) = 8.685$, $P = 0.0015$.

$n = 9$ mice/group. **f** TNF-α levels in the cortical samples. Statistical analysis by one-way ANOVA with Bonferroni's post hoc test. $F(2, 12) = 9.217$, $P = 0.0038$. $n = 5$ mice/group. **g–l** Electrophysiological recordings of sEPSCs and sIPSCs. **h, i** Cumulative fraction analysis of sEPSC of frequency and amplitude ($n = 10$ neurons in the NC group, HHE + S group and HHE + L group). Frequency, statistical analysis by one-way ANOVA with Bonferroni's post hoc test, $F(2, 27) = 7.062$, $P = 0.0034$. **k, l** Cumulative fraction analysis of sIPSC of frequency and amplitude ($n = 10$ neurons in the NC group, HHE + S group and HHE + L group). Total 10 neurons in each recording and 3 mice in each group. **m** Representative image and quantification of western blot. Statistical analysis by one-way ANOVA with Bonferroni's post hoc test. p-PI3K/PI3K, $F(2, 6) = 219.6$, $P < 0.0001$. p-AKT/AKT, $F(2, 6) = 11.62$, $P = 0.0086$. Relative NF-κB protein, $F(2, 6) = 23.93$, $P = 0.0014$. $n = 3$ mice/group. All data are presented as mean values +/– SEM. *$P < 0.05$; **$P < 0.01$; ****$P < 0.0001$. Illustrations created with BioRender.com. Source data are provided as a Source data file.

Our findings suggest that abnormal bile acid metabolism is one possible way in which the gut microbiota imbalance promotes neuroinflammation and neurological impairment. In the HHE group, we detected changes in the abundance of gut microbiota-related metabolites in the serum, and secondary bile acid biosynthesis was the top signalling pathway identified in the enrichment analysis of these altered metabolites. Among them, LCA is an unconjugated bile acid that enters the blood through passive absorption in the colon, and its absorption is highly dependent on the impermeability of the intestinal barrier[46]. HHE-induced gut microbiota dysbiosis (e.g., decreased abundance of *L. murinus*) can impair the permeability of the intestinal barrier, allowing more passive absorption of bile acids such as LCA into serum[47], which subsequently results in a decrease in LCA excretion in faecal samples. After *L. murinus* treatment, the permeability of the intestinal barrier decreased, LCA absorption into blood decreased, and LCA excretion in faecal samples increased. Similarly, the LCA level was significantly increased in the HHE group and was negatively correlated with the abundance of *L. murinus* in the faecal samples. LCA is known to be cytotoxic in rodents, as well as in several human cell types, and it has been reported to be significantly elevated in the serum of patients with anxiety disorders[48]. In addition to LCA, other secondary bile acids were also upregulated in serum, such as taurochenodesoxycholic acid, glycocholic acid, taurodeoxycholic acid, taurine-α-ratcholate sodium salt, and taurocholic acid, were also upregulated in the serum of the HHE group compared to the NC group (Fig. 2j). Among them, LCA is a monohydroxylated secondary bile acid formed from the primary bile acid CDCA and is one of the most hydrophobic natural bile acids[49]. The hydrophobic nature of bile acids allows them to enter cells through sodium-independent transport and passive diffusion[50,51], which allows LCA to enter the blood. Our study revealed that an elevated LCA level played a critical role in causing abnormalities in the HHE group. Using RNA sequencing, RT-PCR and Western blotting of cortical samples, PI3K/AKT/NF-κB signalling was upregulated in the HHE group, and similar results were obtained from the GF-HHE group. Elevated serum LCA levels may cause the liver to produce inflammatory cytokines and impair the BBB, leading to neuroinflammation in the brain. In addition, the increase in the serum LCA concentration was positively correlated with the phosphorylation of PI3K and AKT in cortical samples. Previous studies have shown that the phosphorylation of PI3K/Akt plays an essential role in microglial activation by stimulating NF-κB activity[52], followed by an increase in inflammatory cytokine release[53,54]. Therefore, we speculate that the increase in inflammatory factors in the periphery induced by LCA disrupts the blood–brain barrier and that the increase in neuroinflammation in the brain induced by LCA causes the brain to produce more inflammatory factors in the blood, ultimately resulting in an inflammatory cascade.

Our studies indicate that *L. murinus* administration can fundamentally alleviate anxiety-like behaviours in mice after long-term exposure to a humid heat environmental. Following oral administration, *L. murinus* restored the gut microbiota diversity, downregulated

LCA and TNF-α levels, and decreased PI3K/AKT/NF-κB signalling. Subsequently, the imbalanced E/I ratio of cortical pyramidal neurons and anxiety-like behaviours were efficiently reversed in the HHE group. Interestingly, decreased abundance of *L. murinus* in the faecal samples and upregulation of serum LCA levels had also been identified in the subject population during the humid heat season, and an increasing trend of the level of the proinflammatory cytokine TNF-α was detected in the serum. This finding indicates that a humid heat environment may have a similar effect on humans.

In conclusion, long-term exposure to a humid heat environment is a risk for anxiety disorder directly caused by neuroinflammation and an impaired E/I balance, which are attributed to the downregulation of key gut microbiota and the upregulation of related bile acids, and probiotic administration is a potential therapeutic strategy.

## Limitations of the study
First, neuroinflammation is not a specific cause of anxiety disorders, and whether other types of neurological dysfunction occur in our mouse models needs further study. Second, the subject population showed gut microbiota disturbances and bile acid metabolism abnormalities in the humid heat season that were similar to our mouse models, but long-term cohort studies are required to decode the effects on their neuropsychiatric health.

## Methods
### Animal experiments
All the animal experimental protocols have been pre-approved by the Ethics Committee of Experimental Animals of Jinan University in accordance with Institutional Animal Care and Use Committee guidelines for animal research (Approval code: IACUC-20210528-14). C57BL/6 mice (7 weeks old) were purchased from the Laboratory Animal Centre of Guangdong Province (Guangzhou, China), and assigned to two groups using a computer based random order generator: housing in a normal, conventional environment (22–24 °C and 45–55% humidity; the NC group) or in a humid heat environment (31–33 °C and 91–95% humidity; the HHE group) (12 mice per group, consisting of four cages with 3 mice per cage). GF mice (7 weeks old, C57BL/6 background) were purchased from the Department of Laboratory Animal Science, the First Affiliated Hospital of Sun Yat-sen University (Guangzhou, China), and housed in a GF environment (12 germ-free mice per group, consisting of four cages with 3 mice per cage). GF animal experimental protocols have been pre-approved by the Ethics Committee of Experimental Animals of IEC for Clinical Research and Animal Trials of the First Affiliated Hospital of Sun Yat-sen University with Institutional Animal Care and Use Committee guidelines for animal research (Approval code: [2023]152). For faecal transplantation, stools were collected from mice in the HHE and NC groups and dissolved in 0.01 M phosphate-buffered saline (PBS) at a final concentration of 0.7 g/mL, and the stool (200 µL/day) was transferred to GF mice through oral gavage for 14 consecutive days. All animals were

housed on a 12-h light/dark cycle with *ad libitum* access to food (Jiangsu Xietong Pharmaceutical Bio-engineering Co., Ltd.; 1010009) and water. The behaviour test time was between 08.30 am to 12.30 pm, and the testing order was randomised daily, with each animal tested at a different time each test day.

## 16 S rRNA gene sequencing and analysis

Total microbial genomic DNA was extracted from mouse faecal pellets using the DNeasy PowerSoil Kit (Qiagen, Hilden, Germany), and the V3–V4 regions of bacterial 16 S rRNA genes were amplified using PCR. The PCR products were purified with a Qiagen Gel Extraction Kit (Qiagen, Germany), and sequencing was performed by METWARE Biotechnology Co., Ltd. (Wuhan, China). Paired-end reads were assigned to samples based on their unique barcodes and were truncated by removing the barcode and primer sequences. Paired-end reads were merged using FLASH (v1.2.11, http://ccb.jhu.edu/software/FLASH/), a very fast and accurate analysis tool, which was designed to merge paired-end reads when at least some of the reads overlap the read generated from the opposite end of the same DNA fragment, and the splicing sequences were called raw tags. Quality filtering of the raw tags was performed under specific filtering conditions to obtain the high-quality clean tags according to the QIIME (V1.9.1, http://qiime.org/scripts/split_libraries_fastq.html) quality control process. The tags were compared with the reference database (Silva database https://www.arb-silva.de/) and the UCHIME algorithm (http://www.drive5.com/usearch/manual/uchime_algo.html) to detect chimaeric sequences, after which the chimaeric sequences were removed. Then, the effective tags were obtained. Sequence analysis was performed using Uparse software (Uparse v7.0.1001, http://drive5.com/uparse/). Sequences with ff97% similarity were assigned to the same OTUs. A representative sequence for each OTU was screened for further annotation. Amplicon sequence variants (ASVs) were analysed with Deblur (v1.1.0), which uses error profiles to obtain putative error-free sequences from the Illumina MiSeq and HiSeq sequencing platforms. Multiple sequence alignment was conducted using MAFFT (v7.490, https://mafft.cbrc.jp/alignment/software/) to study the phylogenetic relationships of different OTUs and differences in the dominant species in different samples (groups). OTU abundance information was normalised using a standard sequence number corresponding to the sample with the fewest sequences. Subsequent analyses of alpha diversity and beta diversity were all performed based on these normalised data. The differential abundance of bacterial species was analysed using the multivariate statistical model MaAsLin2.16. Bacterial species with an abundance >0.05% in at least one sample, an adjusted $p < 0.05$ (false discovery rate (FDR)-corrected) and a fold change in abundance >1.5 were considered statistically significant. An alpha diversity analysis was performed to determine the bacterial species complexity in a sample through 6 indices (observed species, Chao1, Shannon, Simpson, ACE, and Good's coverage indices), which were calculated using QIIME and visualised with R software (version 4.1.2). Beta diversity analysis was used to evaluate differences in bacterial species complexity among different samples with principal coordinate analysis (PCoA). Weighted and unweighted UniFrac distances were calculated using QIIME software. Cluster analysis was preceded by PCoA to reduce the dimensionality of the original variables using the stats package and ggplot2 package in R software. PCoA was performed to obtain principal coordinates and visualise complex, multidimensional data, which were displayed using the stats package and ggplot2 package in R software.

## Analysis of human faecal samples

Ethical approval for this study was obtained from the Research Ethics Board of Research Ethics Committee of the First Affiliated Hospital of Jinan University (Guangzhou, China, JNUKY-2022-012). All participants provided signed informed consent before joining the study. Sex has

been consistent with gender information determined by self-reporting, appearance, and identity card. We collected the daily mean, maximum and minimum temperatures as well as the daily relative humidity in 2019–2021 from the Guangzhou weather stations of the China Meteorological Data Service Centre (http://data.cma.cn/). Group the measurement values using statistical methods of time series analysis. The data were used as a reference to determine the start and end dates of each environmental exposure phase. The exposure conditions were classified as: phase 1 (temperature: 22–25 °C; humidity: 30–60%) or phase 2 (temperature: 32–35 °C; humidity: 90–95%). The exposure time at each phase was approximately 4 weeks. Faecal samples were collected from individual participants at the end of phase 1 and at the end of phase 2. The faecal DNA of each subject was extracted according to the QIAamp DNA Stool Mini Kit (QIAGEN, Germany) operating steps. The extracted faecal DNA samples were stored at −20 °C until analysis. The inclusion criteria for participants were as follow: 20–45-year-old males; no history of smoking or stopped smoking for more than 1 year; no incidence of diarrhoea in the last 1 month; no regular dietary habits (no overeating or eating a large amount of spicy or stimulating food within 1 month); and no use of antibacterial drugs in the last 1 month. Faecal samples were collected within 3 min after defecation using sterile stool preservation tubes that contain stool preservation fluid and have a sterile spoon attached to the lid (Tinygene Biologicals, Shanghai, China, Product No. GWF01-A). DNA amplify and the sequencing was done by the METWARE Biotechnology Co., Ltd. (Wuhan, China).

## Faecal microbiota transplantation (FMT)

According to the guidelines for reporting on animal faecal transplantation (GRAFT) studies[55], the methods used for FMT in our study are described below. FMT was achieved by oral gavage of a faecal slurry. The recipient mice had their food removed from the cage for 2 h prior to FMT. The faecal slurry was obtained by pooling faecal pellets from 8–14 donor mice. Immediately, the sample was placed on an anaerobic workbench subjected to the inert gases $N_2$, $H_2$, and $CO_2$ (80%, 5%, and 15%). The pellets were weighed and resuspended in 1 mL of PBS per 300 mg of faeces by vortexing for 1 min. After pelleting larger particles by centrifugation at $500 \times g$ for 5 min, the supernatant was collected for FMT. Each recipient mouse received 200 µl of the faecal slurry by oral gavage once a day for 14 consecutive days.

## Metabolomic analysis

Metabolite extraction, nontargeted LC–MS/MS analysis, and data preprocessing and annotation were performed by the METWARE Company (Wuhan, China). Briefly, 200 µL of serum sample was used for UHPLC–QTOF–MS analysis. Nontargeted LC–MS/MS analysis was performed using a UHPLC system (1290, Agilent Technologies, Santa Clara, CA) with a UPLC BEH Amide column (1.7 µm, 2.1*100 mm; Waters Corporation, Milford, MA) coupled to a Triple-TOF 6600 (Q-TOF, AB Sciex, Redwood City, CA). A triple TOF mass spectrometer was used to acquire MS/MS spectra on an information-dependent basis (IDA) during the LC/MS experiment. The MS raw data files were converted to mzXML format using ProteoWizard and processed with the R package XCMS (version 3.2). The data matrix consisted of the retention time, mass-to-charge ratio, and peak intensity. The R package CAMERA was used for peak annotation after XCMS data processing. The MS2 database was used for metabolite identification, and the metabolomic data were analysed using the MetaboAnalystR R package. Alterations of metabolites were determined using a 2-tailed Mann–Whitney $U$ test, and differences with adjusted $P$ (FDR) values < 0.05 were considered to indicate statistical significance. The associations of differentially abundant bacteria with metabolites were computed using partial Spearman's correlation coefficients, and heatmaps were generated using the Complex Heatmap R package.

KEGG annotation and enrichment analysis, identified metabolites were annotated using KEGG Compound database (http://www.kegg.jp/kegg/compound/), annotated metabolites were then mapped to KEGG Pathway database (http://www.kegg.jp/kegg/pathway.html). Pathways with significantly regulated metabolites mapped to were then fed into MSEA (metabolite sets enrichment analysis), their significance was determined by hypergeometric test's p-values.

## Targeted mass spectrometry assay for LCA

Prior to sacrifice and sample harvest, all mice were observed under their original housing conditions for 1 week; animals were not considered if they exhibited significant signs of serious injury or morbidity (e.g., malocclusion or fight wounds). Upon euthanization in $CO_2$ chambers, blood sera were harvested, snap-frozen, and stored in −80 °C freezer before analysis. Human participants fasted after 7:00 p.m. the day before the examination; 5 mL of venous blood was collected from the elbow from 6:00 a.m. to 9:00 a.m. on the same day. Centrifuge to obtain serum.

Serum samples ($n = 6$ mice/group, $n = 20$ subjects/group. 50 μL) were extracted using a methanol/acetonitrile (v/v = 2:8) mixture (200 μL). Ten microlitres of mixed internal standard solution (1 μg/mL) was added to the extract as an internal standard for quantification. The samples were placed at −20 °C for 10 min to precipitate the proteins, and the supernatant was collected after 10 min of centrifugation (13,800 × g at 4 °C). After evaporation to dryness, the samples were reconstituted in 100 μL of 50% methanol (v/v) for further LC–MS/MS analysis.

Brain samples ($n = 6$ mice/group, 20 mg) were extracted with 200 μL of methanol/acetonitrile(v/v = 2:8) after the samples were ground a ball mill. Ten microlitres of an internal standard mixed solution (1 μg/mL) was added to the extract as an internal standard (IS) for quantification. The samples were incubated at −20 °C for 10 min to precipitate the proteins. After centrifugation for 10 min (13,800 × g, and 4 °C), the supernatant was transferred to clean plastic microtubes. The extracts were evaporated to dryness and reconstituted in 100 μL of 50% methanol (v/v) for LC–MS analysis.

The sample extracts were analysed using an LC-ESI-MS/MS system (UHPLC, ExionLC™ AD, https://sciex.com.cn/; MS, Applied Biosystems 6500 Triple Quadrupole, https://sciex.com.cn/). The analytical conditions were as follows, HPLC: column, Waters ACQUITY UPLC HSS T3 C18 (100 × 2.1 mm i.d.'1.8 μm); solvent system, water with 0.01% acetic acid and 5 mmol/L ammonium acetate (A), acetonitrile with 0.01% acetic acid (B); The gradient was optimised at 5% to 40%B in 0.5 min, then increased to 50% B in 4 min, then increased to 75% B in 3 min, and then 75% to 95% in 2.5 min, washed with 95%B for 2 min, finally ramped back to 5% B (12–14 min); flow rate, 0.35 mL/min; temperature, 40 °C; injection volume: 3 μL. The effluent was alternatively connected to an ESI-triple quadrupole-linear ion trap (QTRAP)-MS.

Linear ion trap (LIT) and triple quadrupole (QQQ) scans were acquired on a triple quadrupole-linear ion trap mass spectrometer (QTRAP), QTRAP® 6500 + LC-MS/MS System, equipped with an ESI Turbo Ion-Spray interface, operating in negative ion mode and controlled by Analyst 1.6.3 software (Sciex). The ESI source operation parameters were as follows: ion source, ESI-; source temperature 550∘C; ion spray voltage (IS) -4500 V; curtain gas (CUR) was set at 35 psi, respectively. Bile acids were analysed using scheduled multiple reaction monitoring (MRM). Data acquisitions were performed using Analyst 1.6.3 software (Sciex). Multiquant 3.0.3 software (Sciex) was used to quantify all metabolites. Mass spectrometer parameters including the declustering potentials (DP) and collision energies (CE) for individual MRM transitions were done with further DP and CE optimisation. A specific set of MRM transitions were monitored for each period according to the metabolites eluted within this period.

## Behavioural tests

For behavioural tests, three people performed the experiment, with the first person modelling the mice according to a randomisation table and marking each mouse. The second person conducted behavioural tests without knowing the specific groups. The collected experimental data were analysed and statistically analysed by a third person.

**Open field test.** Mice were placed in an apparatus (40 × 40 × 30 cm) made of plastic with a white base, and the movement traces were recorded for 10 min using a video camera after individual mice were initially placed in the centre zone. The total distance moved, time spent in the centre zone and number of entries to the centre were calculated using Noldus Observer software (EthoVision 11.0).

**Elevated plus-maze.** Mice were tested in a cross-shaped maze consisting of two open arms (50 × 10 cm), two closed arms (50 × 10 cm), and a central region (10 × 10 cm). Each mouse was placed in the central region of the maze and allowed to explore for 5 min. The time spent in each arm was recorded using Noldus Observer software (EthoVision 11.0).

**Forced swimming test.** Mice were placed in a transparent cylindrical glass container (10 cm in diameter, depth of 22 cm) filled with water (23–25 °C) and a video camera in front of the container. After a 2-min adaptation, the immobility time of each mouse was calculated in a subsequent 4-min recording. Immobility was defined as the absence of any movement of the body of the mouse except for keeping the nose above the water.

**Tail suspension test.** Mice were suspended by their tails using adhesive tape (1 cm from the tail tip), and their reactivity was recorded for 6 min using a video camera. The duration of immobility in the last 4 min was analysed using the EthoVision tracking software programme.

**Sucrose preference test.** One day before the test, the mice were allowed to drink tap water or a 1% (w/v) sucrose solution with an 8-h switch using identical bottles. In the next 24 h, the mice were allowed to drink from 2 bottles: one containing sucrose solution and the other filled with tap water. The sucrose preference was quantified as the ratio of sucrose intake to total fluid intake.

## L. murinus, L. reuteri and Akkermansia muciniphila administration

Adult male mice (7 weeks old) were exposed to a humid heat environment (31–33 °C and 91–95% humidity) for 45 days and then administered L. murinus (BNCC194688, BNCC), L. reuteri (BNCC192190, BNCC) or Akkermansia muciniphila (BNCC341917, BNCC) through oral gavage of live L. murinus, L. reuteri or Akkermansia muciniphila (suspended in physiological saline) at a dose of 108 colony-forming units per day for 14 consecutive days. Mice that received the same volume of physiological saline were considered the controls.

## Electrophysiological recording of acute brain slices

Mice were decapitated after deep anaesthesia with isoflurane, and the brains were immediately isolated and cut into 250-thick slices containing the medial prefrontal cortex (mPFC) using a vibratome (VT1000S, Leica, Germany). The tissue was completely immersed in ice-cold artificial cerebrospinal fluid (ACSF; 126 mM NaCl, 2.5 mM KCl, 1.2 mM NaH2PO4, 10 mM glucose, 26 mM NaHCO3, 2.4 mM CaCl2, 1.2 mM MgCl2, and 295 mM mOsm, pH adjusted to 7.4) with mixed gases (95% $O_2$ and 5% $CO_2$). The slices were then incubated in preheated ACSF (33.5 °C for 30 min) and returned to room temperature for 30 min. The neurons in the mPFC were injected with depolarising current pulses with strengths ranging from ±90 to 300 pA. An

intracellular solution (135 mM K-gluconate, 5 mM KCl, 10 mM HEPES, 0.2 mM EGTA, 4 mM MgATP, 10 mM Na2-phosphocreatine and 0.3 mM NaGTP, pH adjusted to 7.4) was applied to fill the electrodes, and the membrane potential was held at −70 mV. Spontaneous excitatory postsynaptic currents (sEPSCs) were recorded using 20 μM bicuculline (2503/10, Tocris) and a K⁺-based peptide solution at the electrodes. Spontaneous inhibitory postsynaptic currents (sIPSCs) were recorded in 20 μM 2,3-dihydroxy-6-nitro-7-sulfamoyl-benzoquinoxaline (NBQX, 0373/10, Tocris) and 50 μM D-(-)-2-amino-5-phosphonopentanoic acid (D-AP5, 0106/1, Tocris) solutions in flowing ACSF and a KCl-filled pipette (140 mM KCl, 10 mM HEPES, 0.2 mM EGTA, 4 mM MgATP, 10 mM Na2-phosphocreatine and 0.3 mM NaGTP, pH adjusted to 7.4). The recordings were processed using a Multiclamp 700B amplifier (Molecular Devices). The data were discarded if the series resistance was exceeded 20%. The data were analysed using Clampfit 10.0 software (Molecular Devices).

## LCA treatment
Mice were performed a tail vein injection of lithocholic acid (iv; 3 μg/mice/day for 7 days). In parallel, mice were injected with saline (equal volume) as control.

## Histopathology
Colon and liver tissues were fixed with 4% paraformaldehyde (PFA) overnight and embedded in paraffin. The samples were cut into 5-μm sections. The degree of colonic and liver injury was assessed in a blinded manner after performing haematoxylin and eosin (H&E) staining.

## Immunofluorescence staining
After anaesthetisation, the mice were perfused with 4% PFA, and the brains were dissected and postfixed at 4 °C overnight. Brain sections (40 μm thick) were prepared using a sliding microtome (HM450, Leica, Germany) and subjected to immunofluorescence staining using the floating method. The signals were visualised with Alexa Fluor 488- or 594-conjugated secondary antibodies, and the nuclei were counterstained with 4′,6-diamidino-2-phenylindole (DAPI; D3571; Thermo Fisher). Images were captured with a confocal microscope (Zeiss, Germany) and ImageJ2 microscope (Zeiss, Germany).

## Quantitative real-time PCR (RT-PCR)
Total RNA was extracted from the mouse prefrontal cortex using TRIzol reagent (Invitrogen, Carlsbad, CA, USA) according to the manufacturer's instructions, and the RNA concentration was determined using a Nanodrop 2000c spectrophotometer (Thermo, USA). cDNA was synthesised from 1 μg of RNA using a PrimeScript™ RT Reagent Kit with gDNA Eraser (RR047A, Takara, Japan), and the PCR mixture was prepared with TaKaRa TB Green™ Premix Ex Taq™ II (RR820A, Takara, Japan). The cDNA in the mixtures was subjected to PCR using a real-time PCR system (Bio-Rad, USA). The experiments were performed in triplicate, and the relative levels of mRNAs were normalised to the level of GAPDH. The primers were listed in Supplementary Table 1.

Total DNA was isolated and purified from the caecal content using the QIAamp DNA Stool Mini Kit (Germantown, MD). The quality of the DNA samples was assessed using gel electrophoresis and spectrophotometry at 260 and 280 nm. Only DNA samples of sufficient quality were subjected to PCR. Real-time PCR assays of Actinobacteria, Firmicutes and Bacteroidetes were performed as previously described; 16 S rRNA was used as an internal control. The primers were listed in Supplementary Table 1.

## Western blots
The cortical, liver and intestinal tissues were lysed on ice for 30 min in radioimmunoprecipitation assay lysis buffer (RIPA, Thermo Fisher, USA) supplemented with 1 mM protease and phosphatase inhibitor cocktail (Thermo Fisher) and homogenised via ultrasonication for a few seconds. Lysates were clarified via centrifugation at 13,400 × g for 15 min, and protein concentrations were quantified using a Pierce BCA Protein Assay Kit (Thermo Fisher). The samples were subjected to 8% or 10% sodium dodecylsulfate polyacrylamide gel electrophoresis. All gels were transferred to polyvinylidene fluoride membranes by electroblotting. The membranes were blocked with nonfat milk for 2 h and then incubated with primary antibodies overnight at 4 °C. The primary antibodies used are listed in Supplementary Table 1. After washing, the membranes were incubated with secondary antibodies for 2 h, washed, and treated with an enhanced chemiluminescence (ECL) reagent (Merck Millipore). Chemiluminescent signals were detected using a Tanon-5500 chemiluminescent imaging system (Tanon Science and Technology Co., Ltd., China). The signal intensities were analysed using ImageJ software (National Institutes of Health, USA).

## Cytokine analysis
Cytokines in mouse serum and cortical samples were measured using the Bio-Plex 200 System (Bio-Rad, USA) with a Bio-Plex Pro™ Reagent Kit (Cat. No. #M60009RDPD; Bio-Rad, USA), including antibodies against IL-1α, IL-1β, IL-2, IL-3, IL-4, IL-5, IL-6, IL-9, IL-10, IL-12(P40), IL-12(P70), IL-13, IL-17, eotaxin, G-CSF, GM-CSF, IFN-γ, KC, MCP-1, MIP-1α, MIP-1β, RANTES and TNF-α. Cytokine levels in human serum were measured using the MSD MULTI-SPOT Assay System (MSD, USA) with a Proinflammatory Panel 1 (human) Kit (Cat. No. K15049D; MSD, USA), which antibodies against IL-1β, IL-2, IL-4, IL-6, IL-8, IL-10, IL-12(P70), IL-13, IFN-γ and TNF-α. The exclusion criterion was a measured concentration <0.08 μg/ml.

## Enzyme-linked immunosorbent assay (ELISA)
The concentrations of IL-6, IFN-γ and TNF-α in mouse serum, liver and brain samples were determined using ELISA kits (Beijing 4 A Biotech Co., Ltd., China) according to the manufacturer's instructions. Fifty microlitres of each sample was used for detection, and the absorbance was measured at 450 nm. The concentration was calculated from a standard curve drawn using Curve Expert 1.4 software.

## Intestinal permeability assay
Mice were fasted for 4 h and then gavaged with 600 mg/kg FITC-dextran (4000 MW, FD4, Sigma, USA). Four hours later, the mice were sacrificed, and blood was collected to obtain serum. In total, 50 μL serum samples were diluted with 0.01 M PBS, and FITC-dextran concentrations were determined with an Infinite F200 PRO apparatus (TECAN, Switzerland) using an excitation wavelength of 485 nm and emission wavelength of 590 nm by plotting against a calibration curve of known concentrations.

## Measurement of BBB permeability
Mice were injected with 2% Evans blue (Tokyo Chemical Industry, Japan) via the tail vein at a dose of 40 mg/kg. Two hours later, the intravascular Evans blue was removed through cardiac perfusion with 0.01 M PBS. The brains were dissected, and the cerebral hemispheres of each brain were collected. One half of the brain was prepared for sagittal frozen sectioning (20 μm in thickness), and the signal was visualised using a confocal microscope (Zeiss, Germany). The other half of the brain was incubated with formamide (protected from light) at room temperature for 2 days and then centrifuged at 900 × g to collect the supernatant. The levels of Evans blue in the supernatant were measured with a spectrophotometer (at 620 nm).

## RNA sequencing
Total RNA was extracted from fresh cortical samples using TRIzol reagent (Invitrogen, CA, USA), and the quantity and purity of the RNA were evaluated using a Bioanalyzer 2100 and RNA 6000 Nano LabChip Kit (Agilent, CA, USA). High-purity (28 S/18 S > 2.0) and high-integrity (RIN > 8.0) RNA samples were used for cDNA library construction and

RNA sequencing at METWARE Biotechnology Co., Ltd. (Wuhan, China). Briefly, approximately 10 µg of RNA from each sample was subjected to poly(A) mRNA isolation with poly(T) oligo-attached magnetic beads (Invitrogen), and a cDNA library was constructed using an mRNA Seq sample preparation kit (Illumina, San Diego, USA). RNA sequencing was performed on the Illumina sequencing platform (HiSeq 4000, Illumina, USA). Gene expression was calculated based on fragments per kilobase of exon per million fragments mapped (FPKM). The FPKM values were averaged for 3 samples in each group, and DESeq2 was used to identify differentially expressed genes (DEGs) between two groups. Genes with a $P < 0.01$ and a fold change >2 were considered significant. The KEGG database was used to identify pathways enriched in DEGs relative to the whole-genome background. The gene set enrichment analysis (GSEA) algorithm calculates an enrichment score reflecting the degree of overrepresentation at the top or bottom of the ranked list of the genes included in a gene set in a ranked list of all genes present in the RNA sequencing dataset.

## Statistical analysis

Normality, variations, and statistical tests of all datasets were analysed using GraphPad Prism 8. Results were present as means ± SEMs. Statistical analysis of data was performed by two-tailed unpaired $t$-test or non-parametric Mann−Whitney-$U$-test (depending on normal distribution of data). Student's two-tailed $t$-test, the Mann–Whitney $U$ test or multiple $t$-test with the Holm-Sidak method was used for comparisons between two groups. Briefly, adjustment for multiple comparisons was done by the Benjamini−Hochberg correction, and depending on the type of comparison. Microbial β-diversity was determined by fitting models with 16 S profiles as distance-based responses using PERMANOVA and visualised by clustering on an MDS plot. No statistical method was used to predetermine sample size. All behavioural tests were performed by persons who were blinded to the group information. For cytokine analysis in human serum, exclusion criteria if measured concentration <0.08 µg/ml (Data with concentration <0.08 µg/ml are considered invalid).

## Data availability

The authors declare that all data supporting the results of this study are available within the paper and its Supplementary Information. Raw sequence data are deposited at the Genome Sequence Archive (GSA) under accession number PRJCA026649. 16 S rDNA amplicon sequencing and targeted metabolomic generated in this study have been deposited at: https://doi.org/10.6084/m9.figshare.24431887. Raw 16 S rDNA sequence data are also deposited at the GSA under accession number PRJCA026637, PRJCA025700 and PRJCA026953. The relevant metabolomics raw data generated for this study have been deposited in the MetaboLights (accession IDs: MTBLS10320 and MTBLS10326). Source data are provided with this paper.

## Code availability

Software programmes used to analyse the data are either freely or commercially available. All other data relevant to the study are included in the article. Network modules were visualised in standalone GUI software such as Adobe Illustrator (version 2022) and CytoScape (version 3.10.2 for Window 10, Supplementary software).

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

## Acknowledgements

We wish to thank Meizhi Wang for assistance with techniques, and Kwok-Fai So for providing critical comments. This work was supported by the National Natural Science Foundation of China (No. 81830114, No. 82174253, No. 82104707, and No. 82374319), State Key Laboratory of Dampness Syndrome of Chinese Medicine (No. SZ2021KF13), Guangzhou Key Projects of Brain Science and Brain-Like Intelligence Technology (No. 20200730009 and No. 20220600003, L.Z.), Guangdong grant 'Key technologies for treatment of brain disorders (No. 2018B030332001, L.Z.), Programme of Introducing Talents of Discipline to Universities (No. B14036), Guangdong Basic and Applied Basic Research Foundation (No. 2024A1515011809).

## Author contributions

X.C. and L.Z. supervised the study; H.W. and T.W. designed and performed the experiments; H.W. and D.L. performed the bioinformatics analysis; J.W., L.Y., and B.C. participated in the behavioural analysis; Q.Z. participated in the electrophysiological recordings; H.X. participated in the germ-free mouse experiment; L.H. and Y.Q. participated in the experimental design and discussion; H.W. prepared the draft; and L.Z. and X.C. revised the manuscript.

## Competing interests

The authors declare no competing interests.

## Additional information

[1]Department of Neurology and Stroke Center, The First Affiliated Hospital & Clinical Neuroscience Institute of Jinan University, Guangzhou 510632, PR China. [2]Guangdong-Hongkong-Macau CNS Regeneration Institute of Jinan University, Key Laboratory of CNS Regeneration (Ministry of Education), Guangdong Key Laboratory of Non-human Primate Research, Guangzhou 510632, PR China. [3]School of Traditional Chinese Medicine, Jinan University, Guangzhou 510632, PR China. [4]Co-innovation Center of Neuroregeneration, Nantong University, Nantong, Jiangsu, PR China. [5]Neuroscience and Neurorehabilitation Institute, University of Health and Rehabilitation Sciences, Qingdao 266071 Shandong, PR China. [6]Center for Exercise and Brain Science, School of Psychology, Shanghai University of Sport, Shanghai 200438, PR China. [7]These authors contributed equally: Huandi Weng, Li Deng, Tianyuan Wang. ✉e-mail: tlibingzh@jnu.edu.cn; tchenxiaoyin@jnu.edu.cn

