## [Peer Review File · Nature Communications]

REVIEWER COMMENTS

Reviewer #1 (Remarks to the Author):

The gut microbiota impacts multiple aspects of brain function and behaviour via the gut-brain axis, and microbial metabolites are important conduits of this host-microbe dialogue. The gut microbiota also exhibits substantial compositional and functional plasticity in response to a variety of environmental factors. In this study, the authors evaluate the impact of a humid heat environment (HHE) on behaviour, brain function and relevant microbiota-gut-brain axis signalling pathways. Key observations reported include that HHE-induced compositional alterations in the gut microbiota in an animal model is associated with anxiety-like behaviour, an impact potentially mediated by *Lactobacillus murinus* via impaired bile acid metabolism and enhanced neuroinflammation. Similar observations were recorded in human samples harvested during the humid heat season.

This is an intriguing and very comprehensive study with potentially important translational implications. Notable features include the variety of perspectives from which the hypothesis is evaluated, including the use of FMT and the administration of single bacterial strains. The broad range of behavioural and molecular assessments included is also very impressive. I have the following comments:

(1) I am not entirely convinced about the role of lithocholic acid since it is likely that activation of peripheral inflammatory pathways could also produce similar observations, and the authors report both increases of serum lithocholic acid and increased proinflammatory cytokines in the serum in both the animal and humans. It would be important to either demonstrate that there is also an increase in CNS concentrations of this bile acid, or to demonstrate that it is responsible for the barrier function disruption underpinning the neuroinflammatory phenotype.

(2) Germ-free animals are markedly altered at multiple levels of the gut-brain axis, including many of the behavioural and molecular features reported here. It is thus difficult to parse the impact of the HHE-associated microbiota-induced disruption from such underlying neurodevelopmental consequences of growing up with a gut microbiota. An alternative model of microbiota-disruption, such as FMT following antibiotic-induced knockdown, would be necessary to validate the claims made here.

(3) Additional detail is required for the processing of samples for the faecal transplantation study. Were these samples processed in anaerobic conditions and from fresh or frozen samples for example? I refer the authors to the GRAFT guidelines for additional experimental details that should

be reported (Guidelines for reporting on animal fecal transplantation (GRAFT) studies: recommendations from a systematic review of murine transplantation protocols <https://doi.org/10.1080/19490976.2021.1979878>).

(4) The details provided for the processing of the microbiota sequencing is very limited. What databases and bioinformatic pipelines were used?

(5) The term 'Gut flora' is obsolete, please use gut microbiota throughout the manuscript.

Reviewer #2 (Remarks to the Author):

In this manuscript, "Humid heat environment causes anxiety-like disorder through impairing gut microbiota and bile acid metabolism," the authors attempted to elucidate the potential mechanisms by which humid heat environments can cause anxiety disorders. They found a decrease in intestinal *L. murinus* bacteria and an increase in blood lithocholic acid in mice exposed to a humid heat environment and proposed that these could be inflammation-caused in the brain, resulting in anxiety, in a fecal transplantation and *L. murinus* supplementation experiment. This is a very interesting study, but several issues should be addressed to strengthen the paper.

1. The lack of behavioral abnormalities in heat-exposed female animals is a very important result. Did changes in the microbiome occur? Confirmation of the author's findings of reduced *L. murinus* and elevated lithocholic acid should be required.

2. Could the denaturation of the diet by humid heat treatment have affected the food intake, body growth, and gut microbiota of the mice? The preference of mice for humid heat-treated food and its effect on gut microbiome needs to be investigated.

3. How did the authors determine that *L. murinus* was the key bacterium? Could other *L. reuteri* and *Akkermansia* be recovered from behavioral abnormalities in a humid heat environment? A more detailed analysis of the gut microbiota is needed after supplementation of *L. murinus* bacteria. Are there increases in *L. reuteri* and *Akkermansia*?

4. Similarly, why did they conclude that lithocholic acid is key among the bile acid components?

5. Where do they think the inflammatory cytokines in the blood come from? As lithocholic acid causes liver damage, could damage in the liver be the origin of inflammation? Has the liver been examined?

6. Or do they believe that lithocholic acid disrupts the intestinal or brain barrier? The authors would like to present the mechanisms they envisage, from elevated blood lithocholic acid to elevated inflammatory cytokines in the blood and inflammation in the brain.

7. Although it is understood that the source of lithocholic acid is in the gut and that *L. murinus* are responsible for lithocholic acid synthesis, the mechanism for the inverse correlation between blood and fecal lithocholic acid in Fig 6 is not understood. If barrier disruption is the only reason, then all other metabolites would also increase in blood. The mechanism by which secondary bile acid metabolites, including lithocholic acid, characteristically increase in blood needs to be discussed.

Manuscript ID number

NCOMMS-23-54798-T

We sincerely thank the editor and all reviewers for their valuable feedback that we have
used to improve the quality of our manuscript. The reviewer comments are laid out
below in italicized font and specific concerns have been numbered. Our response is
given in normal font and changes/additions to manuscript are given in the blue text.

**RESPONSE TO REVIEWERS' COMMENTS**

The two reviewers raised a number of constructive criticisms and suggestions. To
fully address them, we performed additional experiments as well as implementing
considerable changes to the manuscript. As a result, we believe the manuscript is
much stronger. We wish to take this opportunity to thank the reviewers for their
valuable input. Below, we summarize the reviewers' comments, and describe point-
by-point how we have addressed them.

**Reviewer 1**

● *The gut microbiota impacts multiple aspects of brain function and behaviour via
the gut-brain axis, and microbial metabolites are important conduits of this host-
microbe dialogue. The gut microbiota also exhibits substantial compositional and
functional plasticity in response to a variety of environmental factors. In this study,
the authors evaluate the impact of a humid heat environment (HHE) on behaviour,
brain function and relevant microbiota-gut-brain axis signalling pathways. Key
observations reported include that HHE-induced compositional alterations in the gut
microbiota in an animal model is associated with anxiety-like behaviour, an impact
potentially mediated by *Lactobacillus murinus* via impaired bile acid metabolism and
enhanced neuroinflammation. Similar observations were recorded in human
samples harvested during the humid heat season. This is an intriguing and very
comprehensive study with potentially important translational implications. Notable
features include the variety of perspectives from which the hypothesis is evaluated,
including the use of FMT and the administration of single bacterial strains. The
broad range of behavioural and molecular assessments included is also very
impressive.*

Response: Many thanks for the reviewer's positive comments. We tried our best to
improve our manuscript according to your constructive comments.

● *Point 1: I am not entirely convinced about the role of lithocholic acid since it is*
*likely that activation of peripheral inflammatory pathways could also produce similar*
*observations, and the authors report both increases of serum lithocholic acid and*
*increased proinflammatory cytokines in the serum in both the animal and humans.*
*It would be important to either demonstrate that there is also an increase in CNS*
*concentrations of this bile acid, or to demonstrate that it is responsible for the barrier*
*function disruption underpinning the neuroinflammatory phenotype.*

**Response:** According to the reviewer's comments, we provided the additional
information in the revision as below: Firstly, we measured the concentration of
lithocholic acid (LCA) in the brain in the NC and HHE groups, showing a significant
increase in the HHE group compared to the NC group (new Extended Data Fig. 5).
These additional results have been added to the revision (Result section: Page 8, Line
154) and are also shown as below.

**Extended Data Fig 5. The level of Lithocholic acid increased in the brain.**

LCA concentration measured by targeted mass spectrometry in the brain showed a
significant increase in the HHE group compared to the NC group. **, $P < 0.01$; Student's
*t*-test; $n = 6$ mice/group.

Secondly, to test the impact of LCA on the BBB permeability and neuroinflammation,
 mice in the NC group were subjected to vein injection of LCA or saline for 7
 consecutive days, and we then analyzed the expression levels of the inflammatory
 factors and tight junction proteins (ZO-1, claudin-1 and occludin-1) in the brain (new
 Extended Data Fig. 12a). The results showed that LCA treatment significantly increased
 the expression of TNF- α and IL-6 (new Extended Data Fig. 12b) and downregulated
 the tight junction proteins (new Extended Data Fig. 14c) compared to the mice treated
 by saline. The results suggest that LCA is responsible for the BBB disruption and
 neuroinflammation. These additional results have been added to the revision (Result
 section: Page 12, Line 273-279) and are also shown as below. We briefly discussed the
 potential mechanisms in the revision (Discussion section: Page 17, Line 430-438)

**Extended Data Fig 12. Increased lithocholic acid impairs the BBB impermeability**
 **and promotes neuroinflammation in mice.**

**a** Illustration of the experimental outflow. **b** Mice received vein injection of LCA for 7
 68 days showed an increase of TNF- α and IL-6 detected by ELISA in the brain compared
 to the mice treated by saline. **c** Western blots showed a significant decrease of junction
 proteins including claudin-1, occludin-1 and ZO-1 in the cortical samples in the mice
 treated by LCA compared to those treated by saline. *, $P < 0.05$; **, $P < 0.01$; Student's
 *t*-test; n=6 mice/group for ELISA and mass spectrometry, and n=3 mice/group for

Western blots.

● ***Point 2: Germ-free animals are markedly altered at multiple levels of the gut-***
***brain axis, including many of the behavioural and molecular features reported***
***here. It is thus difficult to parse the impact of the HHE-associated microbiota-***
***induced disruption from such underlying neurodevelopmental consequences of***
***growing up with a gut microbiota. An alternative model of microbiota-disruption,***
***such as FMT following antibiotic-induced knockdown, would be necessary to***
***validate the claims made here.***

**Response:** Thank the reviewer. We tested the impact of FMT using the microbiota-
depleted mice, and the phenotypes in the HHE group was recapitulated as well (please
see new Extended Data Fig. 10). Briefly, mice were treated by an antibiotic cocktail
(ampicillin, 0.25 mg/mL; neomycin, 0.25 mg/mL; metronidazole, 0.25 mg/mL;
vancomycin, 0.125 mg/mL) to deplete the microbiota as described before¹ and then
underwent FMT from either the NC group or HHE group. Mice received FMT from the
HHE group showed the downregulation of tight junction proteins (new Extended Data
Fig. 10a), anxiety-like behaviors (new Extended Data Fig. 10b), excitability increase of
pyramidal neurons in the cortex (new Extended Data Fig. 10c, d), the upregulation of
phosphorylated PI3K and AKT and increase of NF-kB in the brain (new Extended Data
Fig. 10e). These additional results have been added to the revision (Result section: Page
10, Line 216-222) and are shown as below.

**Extended Data Fig. 10. HHE mice-derived FMT results in microbiota-depleted**
 **mouse recapitulating the phenotypes observed in the HHE group.**

Mice were treated by an antibiotic cocktail to deplete gut microbiota and then
 underwent FMT from the HHE group or the NC group, which were short for the FMT-
 HHE group or the FMT-NC group respectively. **a** Western blots of colon samples
 showed a significant increase of claudin-1 and ZO-1 in the FMT-HHE group compared
 to the FMT-NC group (n = 3 mice/group). **b** The time of travelling the central zone in
 the open-field test and staying close arms of elevated plus maze was significantly
 increased in the FMT-HHE group compared to the FMT-NC group (n=12 mice/group).
 **c, d** Electrophysiological recordings in mouse acute brain slices showed a significant
 frequency decrease of pyramidal neuron sEPSC, but no differences of sEPSC amplitude,
 in the FMT-HHE group compared to the FMT-NC group (**c**). There were no differences

of sIPSC recordings in two groups (**d**). Total 10 neurons in each recording and 3 mice
in each group. **e** Western blots of cortical samples showed a significant decrease of
phosphorylated PI3K and AKT1, total NF- κ B proteins in the FMT-HHE group
compared to the FMT-NC group. *, $P < 0.05$; **, $P < 0.01$; Student's t -test; $n = 3$
mice/group.

● **Point 3: Additional detail is required for the processing of samples for the**
**faecal transplantation study. Were these samples processed in anaerobic**
**conditions and from fresh or frozen samples for example? I refer the authors to**
**the GRAFT guidelines for additional experimental details that should be reported**
**(Guidelines for reporting on animal fecal transplantation (GRAFT) studies:**
**recommendations from a systematic review of murine transplantation protocols**
**<https://doi.org/10.1080/19490976.2021.1979878>).**

**Response:** Thanks. In the revision, we provided additional details (Methods section:
Page 21, Line 537-546) of the sample processing for the faecal transplantation study
referred to the guidelines for reporting on animal fecal transplantation (GRAFT) studies.

● **Point 4: The details provided for the processing of the microbiota sequencing**
**is very limited. What databases and bioinformatic pipelines were used?**

**Response:** Thanks. The details have been added in the revision (Methods section; Page
19, Line 485-506). Detailed as follow: Paired-end reads was assigned to samples based
on their unique barcode and truncated by cutting off the barcode and primer sequence.
Paired-end reads were merged using FLASH (v1.2.11,
<http://ccb.jhu.edu/software/FLASH/>), a very fast and accurate analysis tool, which was
designed to merge paired-end reads when at least some of the reads overlap the read

generated from the opposite end of the same DNA fragment, and the splicing sequences
were called raw tags. Quality filtering on the raw tags were performed under specific
filtering conditions to obtain the high-quality clean tags according to the QIIME
(V1.9.1, http://qiime.org/scripts/split_libraries_fastq.html) quality control process. The
tags were compared with the reference database (Silva database [https://www.arb-](https://www.arb-silva.de/)
[silva.de/](https://www.arb-silva.de/)) and using UCHIME Algorithm
(http://www.drive5.com/usearch/manual/uchime_algo.html) to detect chimera
sequences, and then the chimera sequences were removed. Then the Effective Tags
finally obtained. Sequences analysis were performed by Uparse software (Uparse
v7.0.1001, <http://drive5.com/uparse/>). Sequences with ff97% similarity were assigned
to the same OTUs. Representative sequence for each OTU was screened for further
annotation. Amplicon sequence variant (ASV) were analysed by Deblur, which uses
error profiles to obtain putative error-free sequences from Illumina MiSeq and HiSeq
sequencing platforms. In order to study phylogenetic relationship of different OTUs,
and the difference of the dominant species in different samples (groups), multiple
sequence alignment was conducted using the MAFFT (v7.490,
<https://mafft.cbrc.jp/alignment/software/>). OTUs abundance information were
normalized using a standard of sequence number corresponding to the sample with the
least sequences. Subsequent analysis of alpha diversity and beta diversity were all
performed basing on this output normalized data.

● ***Point 5: The term 'Gut flora' is obsolete, please use gut microbiota throughout***
***the manuscript.***

**Response:** Thank you for the suggestion. In the revision, “gut flora” has been replaced
by “gut microbiota”.

**Reviewer 2**

● *In this manuscript, “Humid heat environment causes anxiety-like disorder*
*through impairing gut microbiota and bile acid metabolism,” the authors attempted*
*to elucidate the potential mechanisms by which humid heat environments can cause*
*anxiety disorders. They found a decrease in intestinal L. murinus bacteria and an*
*increase in blood lithocholic acid in mice exposed to a humid heat environment and*
*proposed that these could be inflammation-caused in the brain, resulting in anxiety,*
*in a fecal transplantation and L. murinus supplementation experiment. This is a very*
*interesting study, but several issues should be addressed to strengthen the paper.*

**Response:** We are very grateful for your constructive and helpful comments.

● *Point 1: The lack of behavioral abnormalities in heat-exposed female animals*
*is a very important result. Did changes in the microbiome occur? Confirmation of*
*the author's findings of reduced L. murinus and elevated lithocholic acid should*
*be required.*

**Response:** Thank the reviewer for this important comment. We collected the faecal
samples from female mice and performed 16S rRNA gene sequencing. Although the
clustering of gut microbiota was different in the NC and HHE groups (Attached Fig.
1a), the abundance of *L. murinus* showed no significant differences between two groups
(Attached Fig. 1b). In addition, we compared the expression of LCA in the female mice
between the NC and HHE groups and found that serum LCA was significantly
decreased in the HHE group compared to the NC group (Attached Fig. 1c). The finding
is different from the observation in the male mice in which the abundance of *L. murinus*
was significantly decreased and LCA was significantly increased in the HHE group.
Thus, humid heat environment imposes different effect on the gut microbiota and

metabolism in males and female mice, and the potential mechanisms are required for
further study in the future. In the work, we focused on the study using male mice. As
suggested by the Editor, we changed our title as ‘Humid heat environment causes
anxiety-like disorder through impairing gut microbiota and bile acid metabolism in
male mice’, and the related explanation was also provided in the text.

**Attached Fig. 1. Impact of humid heat environment on gut microbiota and serum**

**LCA in female mice.** Female mouse faecal samples were collected from the HHE and

NC groups for 16S rRNA sequencing, showing the significant differences of microbial

composition in the PCoA (beta diversity) (a), and the abundance of *L. murinus* was

comparable in the HHE and NC groups (b). LCA concentration in serum of female mice

was significantly decreased in the HHE group compared to the NC group (c). **,

$P<0.01$; Student's *t*-test; $n=6$ mice/group.

● **Point 2: Could the denaturation of the diet by humid heat treatment have**
**affected the food intake, body growth, and gut microbiota of the mice? The**
**preference of mice for humid heat-treated food and its effect on gut microbiome**
**needs to be investigated.**

**Response:** Thanks for this important comment. Actually, we replaced the mice with
fresh food every 3 days to avoid the denaturation of the diet caused by humid heat
treatment. We performed additional experiments to confirm whether the diet exposed

in the humid heat environment for 3 days impose an effect on mouse food intake, body
growth, and gut microbiota. Mice received humid heat environment-exposed diet (The
HHD group) or normal diet (the ND group) for 4 weeks, and mouse weight and food
intake were monitored every week showing no differences in the ND and HHD groups
(Attached Fig. 2a, b). At 4 weeks, 16S rRNA gene sequencing of mouse faecal samples
showed that the microbiome composition was comparable in the ND and HHD groups
(Attached Fig. 2c). Thus, the diet is not one important cause to account for the
phenotypes observed in the HHE group.

**Attached Fig. 2. The diet is not the cause of the phenotypes observed in the HHE**
**group.** Mice received humid heat environment-exposed diet (The HHD group) or
normal diet (the ND group) for 4 weeks. There were no differences of their weight (a)
and food intake (b) in two groups. N = 3 cages per group, 6 mice per cage, two-way
repeated ANOVA followed by Bonferroni's multiple comparisons. In addition, the
microbial composition showed comparable in the ND and HDD groups (c; n=6
mice/group).

● ***Point 3: How did the authors determine that was the key bacterium? Could***
***other L. reuteri and Akkermansia be recovered from behavioral abnormalities in***
***a humid heat environment? A more detailed analysis of the gut microbiota is***
***needed after supplementation of L. murinus bacteria. Are there increases in L.***
***reuteri and Akkermansia?***

**Response:** Thank the reviewer. We supposed that *L. murinus* was the key bacterium
to account for the phenotypes in the HHE group, and this conclusion was supported
by the following information. (i) Among ten altered bacterial groups, the decrease of
*L. murinus* was most significant in the HHE group compared to the NC group (Fig.
2f). (ii) Ecological network interaction analysis showed that the reduction of *L.*
*murinus* was synergistically associated with the reduction of protective bacteria (e.g.,
*L. reuteri*), and that *L. murinus* was the dominant species that dominates interactions
and interacted closely with other protective bacteria (Fig. 2g). (iii) The reduction of *L.*
*murinus* abundance was also identified in the GF mice received FMT from the mice in
the HHE group and the human subjects in the humid heat season (Fig. 3d, Extended
Data Fig. 6d). (iv) *L. murinus* administration reversed mouse abnormalities in the
HHE group. (v) Our finding was also in line with the previous reports: *L. murinus*
significantly alleviated the anxiety like behaviors² and possessed the ability to modify
bile acids³.

We agree with the reviewer that other altered bacteria might be also involved in the
abnormalities observed in the HHE group. To test this, we performed the additional
experiments. HHE-treated mice were subjected to the administration of *L. murinus*, *L.*
*reuteri* (the HHE+*L. reuteri* group), *Akkermansia muciniphila* (the HHE+ Akk group),
or saline (the HHE+S group). In the open-field test, the time that mice travelled the
central area was increased in the HHE+L, HHE+*L. reuteri* group, but not in the
HHE+Akk group, compared with HHE+S group (new Extended Data Fig. 14a). In the
elevated plus maze, the time spent in the open arms was increased in the HHE+L
group, HHE+*L. reuteri* group and HHE+Akk group (new Extended Data Fig. 14b).
However, *L. murinus* treatment induced most significant behavioral improvements.
The results have been added to the revision (Result section: Page 13, Line 312-324)
and are shown as below.

**Extended Data Fig. 14. *L. murinus* have the better improvement for HHE-induced**
 **anxiety disorder compared to *L. reuteri* and *Akkermansia muciniphila*.** **a, b** There
 was a significant increase of travelling the central zone of the open field and staying
 open arms of elevated plus maze in the both HHE+L group and HHE+*L. reuteri* group
 compared to the HHE+S group. HHE+Akk group spent more time stay open arms of
 elevated plus maze compared to the HHE+S group. n=6 mice/group; *, $P<0.05$; **,
 $P<0.01$; ***, $P<0.001$; one-way ANOVA and Tukey's multiple comparisons test.

We also collected the faecal samples from the HHE+L and HHE+S groups and
 performed 16S rRNA gene sequencing. In the HHE+L group (treated with *L.*
 *murinus*), the gut microbiota composition in beta diversity was significantly different
 from that in the HHE+S group (treated by saline) (new Extended Data Fig. 15a). The
 abundance of *L. murinus* and *L. reuteri*, but not of *Akkermansia muciniphila*, was
 significantly increased in the HHE+L group compared to the NC group (new
 Extended Data Fig. 15b). These results have been added to the revision (Result
 section: Page 14, Line 327-333) and are shown as below.

**Extended Data Fig. 15. *L. murinus* treatment alters the gut microbiota in the**
 **HHE group.** Mice in the HHE group were treated with *L. murinus* (the HHE+L
 group) or saline (The HHE+S group), and their faecal samples were collected for 16S
 rRNA gene sequencing. **a** Unweighted UniFrac distance-based analysis showed the
 significant differences of microbial composition in the PCoA (beta diversity) in two
 groups. **b** The abundance of *L. murinus* and *L. reuteri*, but not of *Akkermansia*
 *muciniphila* was significantly increased in the HHE+L group compared to the HHE+S
 group. **, $P < 0.01$; ****, $P < 0.0001$; Student's *t*-test; n= 6 mice/group.

● **Point 4: Similarly, why did they conclude that lithocholic acid is key among**
 **the bile acid components?**

**Response:** Thanks for the reviewer's question, and the related question was also
 mentioned by the first reviewer. Our detailed response is as follows: (i) Among the
 altered secondary bile acids in the HHE group (Fig. 2j), lithocholic acid (LCA) showed
 the highest toxicity⁴ and was known to be closely associated with anxiety disorders⁵.
 (ii) The concentration of serum LCA was negatively correlated with the abundance of
 *L. murinus* in faecal sample (Fig. 2l). (iii) Our FMT experiment demonstrated that gut
 microbiota from the HHE group caused the elevation of serum LCA using germ-free
 mice (Fig. 3g). Upregulation of serum LCA was also identified in the human subjects

during humid heat season and accompanied with the decrease of *L. murinus* abundance
in the faecal samples (Fig. 8d, e). (iii) Our additional experiments as described above
showed that LCA was significantly increased in the brain in the HHE group compared
with the NC group, and that LCA treatment caused the permeability increase of the
BBB and neuroinflammation (new Extended Data Fig. 12 a-c). The additional results
have been added to the revision (Result section: Page 12, Line 273-279).

● **Point 5: Where do they think the inflammatory cytokines in the blood come**
**from? As lithocholic acid causes liver damage, could damage in the liver be the**
**origin of inflammation? Has the liver been examined?**

**Response:** To answer the reviewer's question, we did H&E staining of liver sections
and detected the expression of TNF- α and IL-6 in liver samples from the HHE and NC
groups. We found numerous vacuoles (red arrows) indicating lipid deposition and
elevated pro-inflammatory cytokines (TNF- α and IL-6) in the HHE group compared
with the NC group (new Extended Data Fig. 13a, b). The result indicates that the
damaged liver may be one origin of the inflammatory cytokines in the HHE group.
These results have been added to the revision (Result section: Page 12, Line 279-284)
and are shown as below.

**Extended Data Fig. 13. HHE induces mouse liver damage and inflammatory**
**cytokine secretion.**

**a.** H&E staining showed lipid deposition (red arrows) in the HHE group but rarely in
the NC group. **b.** ELISA of liver samples showed a significant increase of TNF- α and
IL-6 in the HHE group compared to the NC group. *, $P < 0.05$; Student's t -test; $n = 4$
mice/group.

● ***Point 6: Or do they believe that lithocholic acid disrupts the intestinal or brain***
***barrier? The authors would like to present the mechanisms they envisage, from***
***elevated blood lithocholic acid to elevated inflammatory cytokines in the blood***
***and inflammation in the brain.***

**Response:** We agree with the reviewer that LCA may disrupt the brain and/or intestinal
barrier. Our additional experiment as described above demonstrated that LCA treatment
resulted in downregulation of junction proteins (ZO-1, Occludin-1 and Claudin-1) and
upregulation of pro-inflammatory cytokines (TNF- α and IL-6) in the brain (new
Extended Data Fig. 12). In addition, mice in the HHE group treated by colestyramine
(a bile acid binding resin) to reduce bile acid accumulation in the blood showed a
significant decrease of inflammatory factors (TNF- α and IL-6) in the serum and brain
(Attached Fig. 3a, b). This finding is in line with the previous reports⁶. We briefly
discussed the potential mechanisms in the revision (Discussion section: Page 17, Line
430-438) as: Elevated serum LCA may cause the liver to produce inflammatory
cytokines and impair the BBB, leading to neuroinflammation in the brain. In addition,
the upregulation of serum LCA was positively correlated with the phosphorylation of
PI3K and AKT in the cortical samples. Previous studies have shown that
phosphorylation of PI3K/Akt plays an essential role in microglial activation by
stimulating NF- κ B activity⁷, following with the increase of inflammatory cytokines
release⁸⁻⁹. Therefore, we speculate that the inflammatory factors increase in peripheral

induced by LCA disrupts the blood-brain barrier, and the neuroinflammation increased
in the brain induced by LCA causes the brain to produce more inflammatory factors
into the blood, finally resulting inflammatory cascading response.

**Attached Fig. 3. Colestyramine treatment inhibits elevation of TNF- α and IL-6 in**
**the HHE group.** Mice in the HHE group were treated with colestyramine (the HHE+C
group) or saline (the HHE+S group), and blood and cortical samples were collected for
ELISA. The samples from the NC group were used as the reference. In the serum, the
expression of TNF- α and IL-6 was higher in the HHE+S group than the NC group and
the HHE+C group (a), and the similar changes were found in the cortical samples (b).
*, $P < 0.05$; **, $P < 0.01$; one-way ANOVA and Tukey's multiple comparisons test; $n = 6$
mice/group.

● **Point 7: Although it is understood that the source of lithocholic acid is in the**
**gut and that *L. murinus* are responsible for lithocholic acid synthesis, the**
**mechanism for the inverse correlation between blood and fecal lithocholic acid in**
**Fig 6 is not understood. If barrier disruption is the only reason, then all other**
**metabolites would also increase in blood. The mechanism by which secondary bile**
**acid metabolites, including lithocholic acid, characteristically increase in blood**
**needs to be discussed.**

**Response:** We found an inverse change of LCA levels in the serum and fecal samples
after *L. murinus* treatment of mice in the HHE group (see Fig. 6c, d). LCA is one
unconjugated bile acid and enters the blood through passive absorption in the colon,
and the absorption is highly dependent on the impermeability of the intestinal barrier¹⁰.
HHE-induced gut microbiota dysbiosis (e.g., *L. murinus* decrease) could impair the
permeability of the intestinal barrier allowing more passive absorption of bile acids
such as LCA into serum¹¹, which subsequently resulted in the decrease of LCA excretion
in the fecal samples. After *L. murinus* treatment, the permeability of the intestinal
barrier was reduced; LCA absorption into blood was decreased and LCA excretion in
fecal samples was increased. In line with this, we found that the abundance of *L.*
*murinus* was negatively correlated with LCA concentration in serum (Fig. 2l). We added
a brief explanation to the revision (Discussion section: Page 16, Line 408-415).

In addition to LCA, other secondary bile acids were also upregulated in serum such
as taurochenodesoxycholic acid, glycocholic acid, taurodeoxycholic acid, taurine- α -
ratcholate sodium salt, and taurocholic acid in the HHE group compared to the NC
group (Fig. 2j). Among them, LCA is a monohydroxylated secondary bile acid formed
from the primary bile acid CDCA and is one of the most hydrophobic natural bile acids¹².
Hydrophobic nature of bile acids allows them to enter cells through sodium-
independent transporter and passive diffusion¹³⁻¹⁴, and this confers LCA with
predominance to enter the blood. Our study demonstrated that elevated LCA played a
critical role in causing the abnormalities in the HHE group. We added a brief
explanation to the revision (Discussion section: Page 17, Line 419-427)

**References**

- 1. Zhao, Q.; Dai, M. Y.; Huang, R. Y.; Duan, J. Y.; Zhang, T.; Bao, W. M.; Zhang, J.
Y.; Gui, S. Q.; Xia, S. M.; Dai, C. T.; Tang, Y. M.; Gonzalez, F. J.; Li, F., *Parabacteroides*
*distasonis* ameliorates hepatic fibrosis potentially via modulating intestinal bile acid

- metabolism and hepatocyte pyroptosis in male mice. *Nat Commun.* **14** (1), 1829 (2023).
- 2. Li, L. F.; Zou, H. W.; Song, B. L.; Wang, Y.; Jiang, Y.; Li, Z. L.; Niu, Q. H.; Liu, Y.
399 J., Increased Lactobacillus Abundance Contributes to Stress Resilience in Mice
Exposed to Chronic Social Defeat Stress. *Neuroendocrinology.* **113** (5), 563-576 (2023).
- 3. Jukes, C. A.; Ijaz, U. Z.; Buckley, A.; Spencer, J.; Irvine, J.; Candlish, D.; Li, J. V.;
Marchesi, J. R.; Douce, G., Bile salt metabolism is not the only factor contributing to
Clostridioides (Clostridium) difficile disease severity in the murine model of disease.
*Gut Microbes.* **11** (3), 481-496 (2020).
- 4. Holtmann, T. M.; Inzaugarat, M. E.; Knorr, J.; Geisler, L.; Schulz, M.; Bieghs, V.;
Frissen, M.; Feldstein, A. E.; Tacke, F.; Trautwein, C.; Wree, A., Bile Acids Activate
NLRP3 Inflammasome, Promoting Murine Liver Inflammation or Fibrosis in a Cell
Type-Specific Manner. *Cells.* **10** (10) (2021).
- 5. MahmoudianDehkordi, S.; Bhattacharyya, S.; Brydges, C. R.; Jia, W.; Fiehn, O.;
Rush, A. J.; Dunlop, B. W.; Kaddurah-Daouk, R., Gut Microbiome-Linked Metabolites
in the Pathobiology of Major Depression With or Without Anxiety-A Role for Bile
Acids. *Front Neurosci.* **16**, 937906 (2022).
- 6. McMillin, M.; Frampton, G.; Grant, S.; Khan, S.; Diocares, J.; Petrescu, A.; Wyatt,
414 A.; Kain, J.; Jefferson, B.; DeMorrow, S., Bile Acid-Mediated Sphingosine-1-
415 Phosphate Receptor 2 Signaling Promotes Neuroinflammation during Hepatic
Encephalopathy in Mice. *Frontiers in cellular neuroscience.* **11**, 191 (2017).
- 7. Jayasooriya, R. G.; Lee, K. T.; Kang, C. H.; Dilshara, M. G.; Lee, H. J.; Choi, Y.
H.; Choi, I. W.; Kim, G. Y., Isobutyrylshikonin inhibits lipopolysaccharide-induced
nitric oxide and prostaglandin E2 production in BV2 microglial cells by suppressing
the PI3K/Akt-mediated nuclear transcription factor- κ B pathway. *Nutrition research*
*(New York, N.Y.).* **34** (12), 1111-9 (2014).
- 8. Cianciulli, A.; Calvello, R.; Porro, C.; Trotta, T.; Salvatore, R.; Panaro, M. A.,
PI3k/Akt signalling pathway plays a crucial role in the anti-inflammatory effects of

curcumin in LPS-activated microglia. *International immunopharmacology*. **36**, 282-
290 (2016).

9. Tarassishin, L.; Suh, H. S.; Lee, S. C., Interferon regulatory factor 3 plays an anti-
inflammatory role in microglia by activating the PI3K/Akt pathway. *Journal of*
*neuroinflammation*. **8**, 187 (2011).

10. Lennernäs, H., Intestinal permeability and its relevance for absorption and
elimination. *Xenobiotica; the fate of foreign compounds in biological systems*. **37** (10-
11), 1015-51 (2007).

11. Collins, S. L.; Stine, J. G.; Bisanz, J. E.; Okafor, C. D.; Patterson, A. D., Bile acids
and the gut microbiota: metabolic interactions and impacts on disease. *Nature reviews*.
*Microbiology*. **21** (4), 236-247 (2023).

12. Lucangioli, S. E.; Carducci, C. N.; Tripodi, V. P.; Kenndler, E., Retention of bile
salts in micellar electrokinetic chromatography: relation of capacity factor to octanol-
water partition coefficient and critical micellar concentration. *Journal of*
*chromatography. B, Biomedical sciences and applications*. **765** (2), 113-20 (2001).

13. Van Dyke, R. W.; Stephens, J. E.; Scharschmidt, B. F., Bile acid transport in
cultured rat hepatocytes. *The American journal of physiology* 1982, **243** (6), G484-92.

14. Aldini, R.; Roda, A.; Montagnani, M.; Cerrè, C.; Pellicciari, R.; Roda, E.,
Relationship between structure and intestinal absorption of bile acids with a steroid or
side-chain modification. *Steroids*. **61** (10), 590-7 (1996).

REVIEWERS' COMMENTS

Reviewer #1 (Remarks to the Author):

The authors have been very responsive to the initial recommendations. This includes the addition of new experimental data which supports the main thrust of the original paper. I have no further comments.

Reviewer #2 (Remarks to the Author):

My comments are appropriately addressed, thus I have no further remarks.